# Heterogeneous thermal tolerance of dominant Andean montane tree species
Zorayda Restrepo [1,2,3] ✉, Sebastián González-Caro[4], Iain P. Hartley [4], Juan Camilo Villegas[3],
Patrick Meir[5,6], Adriana Sanchez[7], Daniel Ruiz Carrascal [8] & Lina M. Mercado [4,9] ✉

In tropical montane forests, the Earth's largest biodiversity hotspots, there is increasing evidence that climate warming is resulting in montane species being displaced by their lowland counterparts. However, the drivers of these changes are poorly understood. Across a large elevation gradient in the Colombian Andes, we established three experimental plantations of 15 dominant tree species including both naturally occurring montane and lowland species and measured their survival and growth. Here we show that 55% of the studied montane species maintained growth at their survival's hottest temperature with the remaining 45% being intolerant to such levels of warming, declining their growth, while lowland species benefited strongly from the warmest temperatures. Our findings suggest that the direct negative effects of warming and increased competition of montane species with lowland species are promoting increased homogeneity in community composition, resulting in reduced biodiversity.

The tropical Andes are among the most biodiverse regions in the world[1–3]. However, its diversity is at risk due to climate change and habitat loss[1,4], which may affect their functioning and capacity to provide ecosystem services, key for the people in this and adjacent regions[5,6]. The impacts of climate change are a substantial in the northern South America, where both monthly mean and maximum temperatures have increased between 0.6 and 2.4 °C and between 1.2 and 6.6 °C respectively, during the 1950–2010[4,7–9] period. Future projections in this region correspond to 4.5 °C rise in the median temperature by 2100 (with 2.6 and 6.6 °C as the 5[th] and 95[th] percentiles) relative to present day following SSP5-8.5[10] Similar future temperature projections are expected in other tropical montane regions such as in Central Africa but are ~1 °C larger than projections for montane forest in central America. The expected temperature increase across the tropical Andes is elevation-dependent[4] which amplifies the difference between minimum and maximum temperature both diurnally and annually, as well as the frequency of heatwaves[4,11]. In contrast, observed annual precipitation trends do not show a homogeneous pattern across the Andes during the period 1964-2008[4,12], varying between –4% and +4% per decade relative to mean annual precipitation. Large uncertainties remain in future precipitation projections over the Andean region[4,11–13]. It is expected that the predicted systematic temperature rise will affect natural ecosystems across the

tropical Andes however the extent and nature of the of the impacts remains understudied.

There is evidence of strong effects of global warming on tropical tree species, including shifts in their geographical range with consequences for population stability and community composition[14–17]. Observed shifts in tropical tree community composition are consistent with the idea that trees respond to changes in temperature by tracking the range of environmental temperatures within which a species can survive i.e. tracking their thermal range[17,18]. A species' thermal distribution comprises the range of temperatures at which the species is found; within this range, the temperature at which the species grows best is known as their thermal optimum, $T_{opt}$ (Fig. 1). We define the cold and warm portions of a species thermal range as the variation in minimum and maximum temperatures experienced by the species, respectively. The hot extreme of a species thermal range can be defined as the temperature above the 75[th] percentile of the maximum temperature experienced by the species, and their cold extreme as the minimum temperature below the 25[th] percentile of the minimum temperature experienced by the species. The upward movement of species in montane environments from the warm lowlands to cooler uplands produces a reconfiguration of species communities which has been termed thermophilisation[15]: warm affiliated lowland foothill thermophilic species,

[1]Grupo GiGA, Escuela Ambiental, Facultad de ingeniería, Universidad de Antioquia, Medellín, Colombia. [2]Grupo de Servicios ecosistémicos y Cambio Climático, Corporación COL-TREE, Medellín, Colombia. [3]Grupo de Investigación en Ecología Aplicada, Escuela Ambiental, Facultad de Ingeniería, Universidad de Antioquia, Medellín, Colombia. [4]Geography, Faculty of Environment, Science and Economy, University of Exeter, Exeter, UK. [5]School of Geosciences, University of Edinburgh, Edinburgh, UK. [6]Research School of Biology, Australian National University, Canberra, Australia. [7]Departamento de Biología, Facultad de Ciencias Naturales, Universidad del Rosario, Bogotá, D.C., Colombia. [8]Innovation and Technological Development Directorate, Eafit University, Medellín, Colombia. [9]UK Centre for Ecology & Hydrology, Wallingford, UK. ✉e-mail: corporacioncoltree@gmail.com; l.mercado@exeter.ac.uk

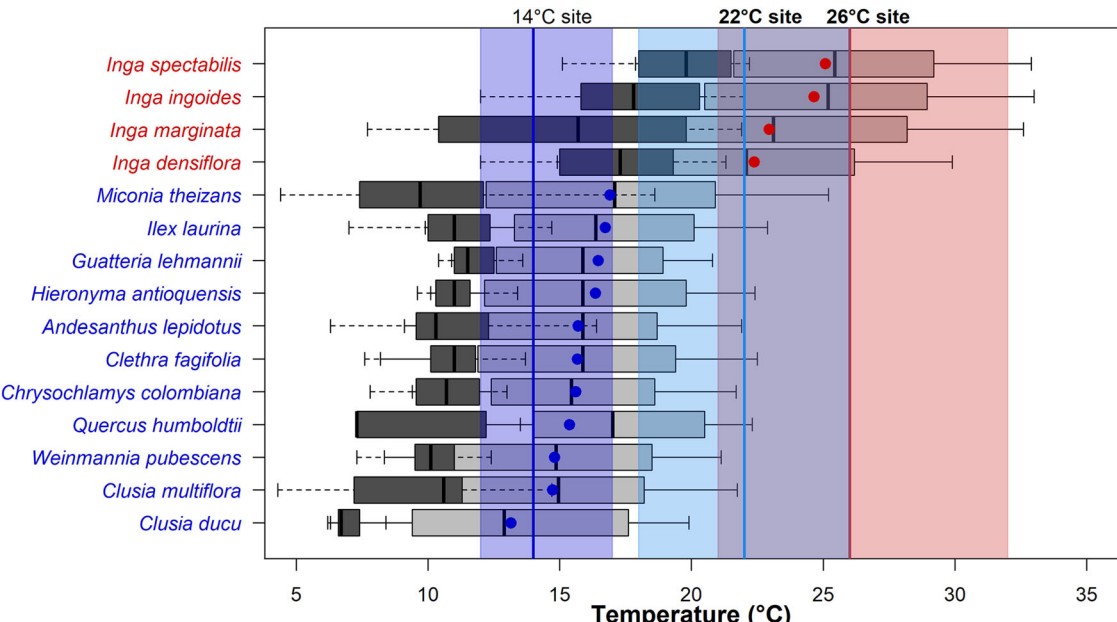

**Fig. 1 | Thermal distribution of study species and range of measured air temperatures at each experimental site.** Boxplots show the variation between the 10th and the 25th percentile of the minimum temperature, $T_{min}$ (dark grey boxes) and between the 25th and the 90th percentile of the maximum temperature $T_{max}$ (light grey boxes) from locations where species were recorded in the BIEN database, representing the cold and warm portion for each species thermal range. $T_{min}$ and $T_{max}$ correspond to the average temperature of the coldest and warmest month respectively during the 1970–2000 period from the WorldClim V.2. dataset[19] for each species record. Blue and red dots represent $T_{opt}$ of each species. Montane and lowland species are in blue and red fonts, respectively. To minimise temperature bias due to geographic occurrence errors, 5% of the data from each tail (5% and 95%) were removed. Coloured vertical polygons represent the thermal environment measured at experimental sites during the period October 1st, 2019, to January 31st, 2022. Sites MAT are represented with vertical coloured lines and correspond to 14 °C, 22 °C, and 26 °C with the lower and upper thermal limits represented by the 10th and 90th percentiles respectively. Note that the 90th percentile of the 22 °C site coincides with the MAT of the 26 °C site.

hereafter termed lowland species, are increasing in abundance across elevations relative to highland cold affiliated montane species, hereafter termed montane species. Observed directional shifts in species composition over time detected on forest plots[14,15,19,20] provide evidence of thermophilisation on tropical montane tree communities in the Andes (reported in Colombia, Peru, Ecuador and Argentina[14,15]), in Afromontane forests (reported in Rwanda, Uganda, Democratic Republic of Congo and Tanzania[20]) and in Central America (reported in Costa Rica[21] and Jamaica[22]). Thermophilisation in Andean forest is consistent with concurrent warming in the region and is caused by (1) increased abundance of lowland species in their upper limit of elevational range which coincides with the cold extreme of the thermal range, expanding their elevational range[14,15,20] and (2) increased mortality of montane species in their lower limit of their elevational ranges (i.e. the range of elevations within which species can survive) which coincide with the hot extremes of their thermal ranges, leading to contractions of their elevational range[19]. However, to date it is not known whether the loss of montane species is due to direct negative impacts of climate warming on tree growth, or whether losses are driven predominantly by increased competition with lowland species. Such understanding is crucial for predicting future rates of change, as well as for planning conservation programmes. Furthermore, although observed species compositional shifts in the tropical Andes support thermophilisation, such change in species composition is heterogeneous across elevations[14,15]. This is expected to be due to differences in species level responses global warming[14,19]. Therefore, understanding species-level responses will enhance our knowledge of the resilience of tropical montane ecosystems to global warming.

According to Shelford's law[23], plant performance is limited by any deficit or excess of environmental conditions or resources (e.g., temperature), leading to a gradual reduction in performance from optimal environment conditions at which a plant grows best to extreme conditions under which a plants performs poorly[24,25]. However, responses of plant performance to extreme cold and to extreme high temperatures may differ due to contrasting metabolic constraints (e.g., chilling vs. heating). The lower limit of a species' thermal distribution is considered a strong limiting factor to plant performance because cool conditions reduce metabolic rates[26], and thus growth. Conversely, the upper range and hot extreme of a species' thermal distribution may abruptly restrict enzymatic activity[27,28]. Therefore, the displacement of a species from its thermal optimum towards the species hot extreme could thus lead to a steeper reduction in tree growth compared to species displacement towards their cold extreme. Quantifying the range of temperatures within which a species can survive (i.e. species thermal range) is complex, it can nevertheless be estimated using herbarium records of temperature variation across species geographical range[29]. From such thermal distributions is plausible to determine the thermal optimum $T_{opt}$ (mean of species' thermal distribution), the minimum and the maximum temperatures experienced by each species. Species responses to changes in temperature are influenced by adaptations to the climate where they grow best, i.e. their thermal affinities[30,31], leading to varying responses under species cold and warm portions of their thermal ranges including respective extremes.

Transplant experiments along elevational gradients provide ideal natural thermal variations to investigate species level responses to warming[32–35]. To overcome differences in soil conditions, pot experiments have been used[35]. This approach has limitations including plant size which can constrain photosynthesis and thus plant growth[36]. Other approaches used to eliminate variations in soil nutrients in transplant experiments in elevation gradients consist in fertilising all trees at all elevations to similar levels of nutrition[19]. However, this approach does not deal with possible differences in soil physical conditions across experimental sites which might affect plant performance. A more logistically challenging method to overcome differences in soil type is to combine seedling/sapling with soil transplant[35], which removes soil conditions as variable of influence on tree performance. Inclusion of a fertiliser treatment as part of such transplant experiment facilitates evaluation of nutrient constraints on tree

performance. Overall, this methodology combined with irrigation and continuous in situ meteorological monitoring allows evaluation of direct and indirect effects of temperature on plant performance in isolation from water and soil nutrient effects during the period that plants grow under common soils at all experimental sites.

Here we evaluated the sensitivity of montane and lowland tree species to changes in temperature. We used a transplant experiment of 15 species (11 montane and 4 lowland) at three experimental sites (under common irrigation, soil texture and nutrients, in open areas planted 2.5 m apart to eliminate competition) in an elevation gradient in the Colombian Andes with mean annual temperature (MAT) of 14 °C, 22 °C, 26 °C (Supplementary Fig. 1). To further our understanding of the mechanisms underpinning forest compositional change, here we tested (1) whether dominant montane species are able to survive and grow at their hot extreme and beyond this hot thermal limit, (2) whether growth and survival of dominant lowland species is higher than for the dominant montane counterparts under the hot extreme of montane species thermal range and (3) under the cold extreme of lowland species thermal range. We hypothesised that survival and growth of all species would decrease when growing away from their thermal optimum and that plants growing under the cold extreme of their thermal range would have better performance in terms of growth and survival than plants growing under their hot extreme. We evaluated species survival and growth across the three experimental sites during a three-year period and found a high variability in species responses to changes in thermal environment. Our analysis demonstrates that survival and growth rates decreased with warming in

montane species but increased in all lowland species with cooling. Nine of the 11 studied montane species grew best at the site close to their thermal optimum (14 °C), and although they were able to survive and grow at their hot extreme (22 °C), their growth rates were lower under those conditions. None of the montane species survived when exposed to temperatures beyond the hot extreme of their thermal distribution (26 °C). In contrast, lowland species exhibited high survival when growing at the cold extreme (14 °C) (Fig. 1) of their thermal range, though their growth rates were significantly reduced compared to growth at their $T_{opt}$. Our results highlight the high variability of responses to temperature across Andean tree species.

## Results

After three years from planting, survival of all species was highest, as expected, when growing at the site with MAT closest to their computed $T_{opt}$; survival rates of montane species varied between 65 and 100%, and between 78 and 100% for lowland species. When growing under temperatures away from species $T_{opt}$, montane species showed an overall negative effect of warming on tree survival based on the Cox Proportional Hazard regression[37,38]: none of the species survived when growing at temperatures outside the hot extreme of their thermal distribution (i.e. 26 °C MAT) and 45% of the originally planted montane trees at their hot extreme (22 °C MAT) survived. However, species level responses varied, and nine out of eleven montane species survived at the 22 °C site (Fig. 2e–o). Specifically, two species (*Quercus humboldtii* and *Ilex laurina*) showed high survival rates (96% and 58%, respectively), four species (*Clusia multiflora, Miconia*

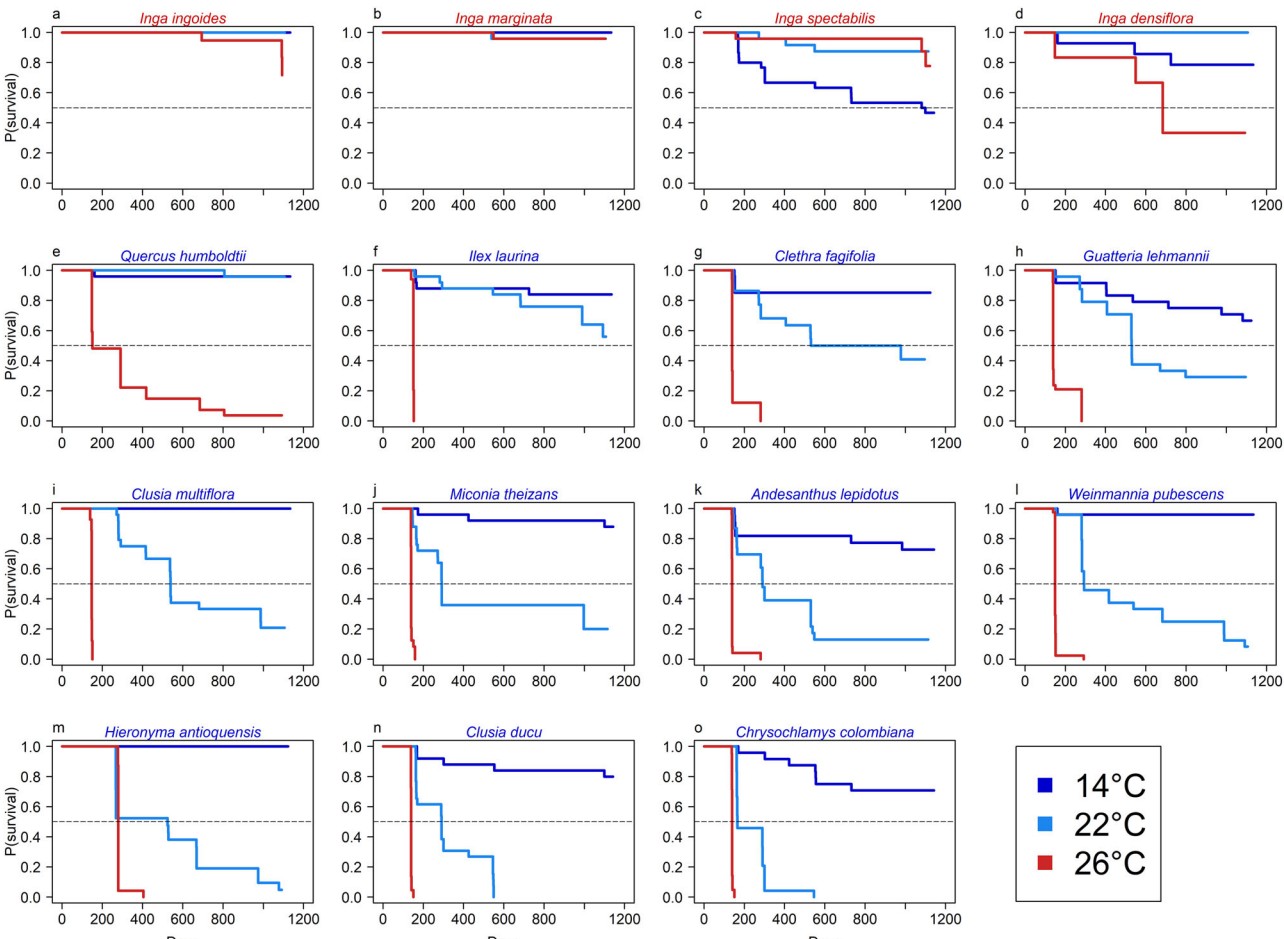

**Fig. 2 | Tree survival responses to temperature.** The probability (P) of individual trees surviving on a particular day for each species from the day of planting (Day 1) at each site (differentiated by site MAT: 14, 22 and 26 °C). Coloured lines are the Cox proportional hazard function per site and x-axes are days since planting. Names of lowland and montane species are in red and blue fonts, respectively. Dashed black horizontal lines represent the 0.5 probability of survival. Panels (**a**) to (**o**) correspond to the probability of survival for each of the study species (named above each panel) at each experimental site during the study period.

*theizans*, *Clethra fagifolia* and *Guatteria lehmannii*) showed intermediate survival rates (between 20% and 40%), three species (*Andesanthus lepidotus*, *Weinmannia pubescens* and *Hieronyma antioquensis*) had very low survival rates (lower than 20%) and two species did not survive at all (*Clusia ducu* and *Chrysochlamys colombiana*). Interestingly, the survival of the latter two species at the site closest to their $T_{opt}$ was the highest among all montane species at that site (71–88%). In contrast, when exposed to the cold portion of their thermal range (14 °C MAT), lowland species generally had survival rates higher than 87% at 22 °C (two species) and above 75% for three out of the four species (Fig. 2a–d).

In terms of relative tree growth rate (RGR, Eq. (1) in "Methods"), overall, all species decreased RGR when growing away from their $T_{opt}$: montane species decreased RGR in response to warming (F = 5.6, $p$ = <0.001) and lowland species decreased RGR in response to cooling (F = 2.3, $p$ = 0.0207) (Fig. 3a–d). However, there were variations across species: five out of the nine montane species (55%) that survived at their hot extreme (22°C MAT), did not show significant warming effects on RGR (i.e., maintained RGR at $T_{opt}$, 14°C MAT and at 22 °C MAT t = -

0.99 $p$ = 0.324), while the remaining four montane species (45%) showed declines in RGR at this temperature (Fig. 3e–m). Of the four lowland species, two maintained the same RGR when growing at 22 °C and 26 °C MAT (Fig. 3a–d). A comparison of RGR of montane and lowland species at a common growth temperature of 22 °C MAT (hot extreme for montane species and within portions of the cold and warm range of lowland species), demonstrates that at this temperature lowland species grow faster (lowland mean RGR = 0.67 ± 0.16 mm.mm$^{-1}$ year$^{-1}$; montane mean RGR = 0.47 ± 0.19 mm.mm$^{-1}$ year$^{-1}$; t = −3.89; $p$ < 0.001). However, at 14 °C MAT (closest to $T_{opt}$ of montane species and cold extreme of three lowland species), montane trees grow faster (lowland mean RGR = 0.42 ± 0.18 mm.mm$^{-1}$ year$^{-1}$; montane mean RGR = 0.56 ± 0.16 mm.mm$^{-1}$ year$^{-1}$; t = 4.07; $p$ < 0.001) (Fig. 3). Species level average tree size after three years of planting and growth rates over time are reported in Supplementary Table 1 and Supplementary Fig. 2, respectively.

We can predict the magnitude of the observed growth responses to changes in temperature for all species when we separate the lowland and

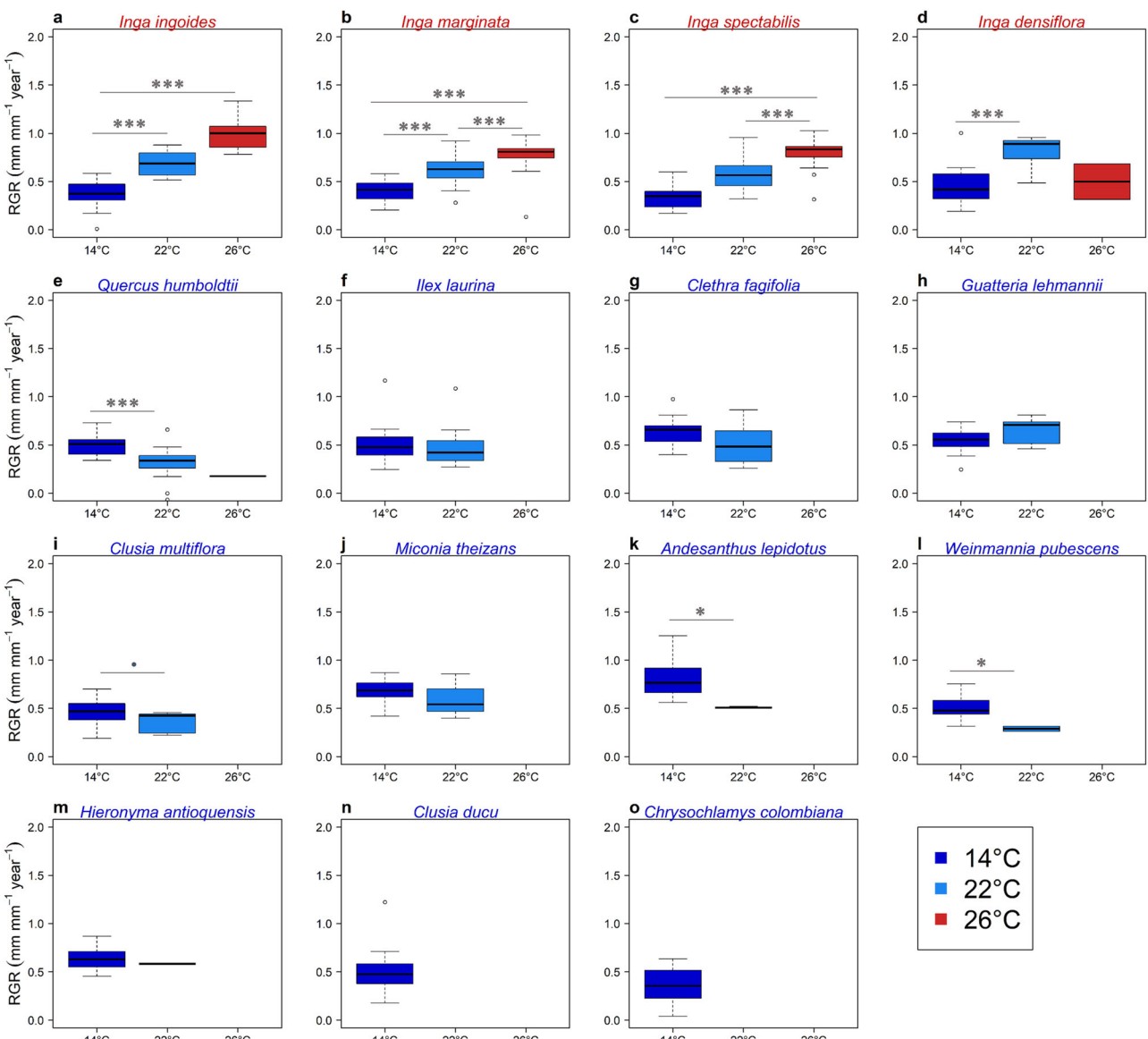

**Fig. 3 | Relative Growth Rate (RGR) responses to experimental site MAT.** RGR indicates the increase in diameter during three years after planting at each experimental site (denoted by site MAT: 14, 22 and 26 °C each with a different colour). Names of lowland and montane species are in red and blue fonts, respectively.

Significant differences among sites are obtained by multi-comparison Tukey tests: *$P$ < 0.05; **$P$ < 466 0.01; ***$P$ < 0.001. Panels (**a**) to (**o**) correspond to responses of relative growth rate of each of the study species (named above each panel) to each experimental site MAT during the study period.

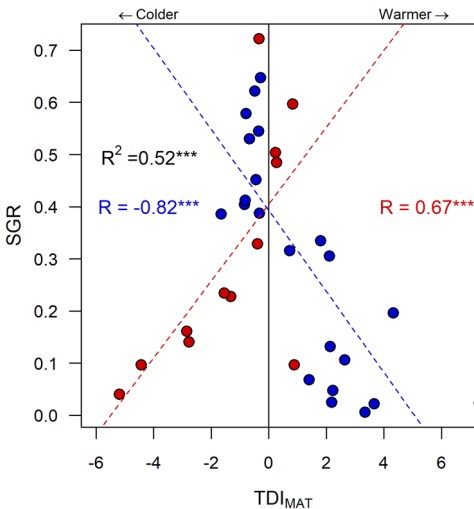

**Fig. 4 | Scaled growth rate (SGR) at species level decreases with increasing thermal displacement from thermal optimum ($TDI_{MAT}$) relative to experimental site MAT.** TDI was estimated using MAT ($TDI_{MAT}$). A high absolute value of $TDI_{MAT}$ indicates a high thermal displacement from species $T_{opt}$ and is due to either a large difference between species $T_{opt}$ and site MAT, a low standard deviation of temperature across a species range, or both. A negative value of TDI means species $T_{opt}$ is larger than the site MAT, and a positive value indicates that species $T_{opt}$ is lower than the site MAT. Data for montane and lowland species are shown in red and blue, respectively. $R^2$ values for the regression between SGR and TDI and significance are shown. Pearson r and significance for each dashed line representing a linear model fit correspond to montane and lowland species.

montane groups. Specifically, we found a strong relationship between scaled growth rate (SGR) and thermal displacement index (TDI) which represents the thermal displacement from the species $T_{opt}$, (Fig. 4). TDI indicates how far above or below species $T_{opt}$ the species has been planted. SGR is highest at sites with MAT (or $MAT^{90th}$ or $MAT^{10th}$ percentiles) closest to the species $T_{opt}$, which corresponds to a low TDI, and it decreases with increasing temperature displacement from $T_{opt}$. Significant relationships were found for both groups of species when estimating TDI with MAT ($TDI_{MAT}$), MAT 90th percentile ($TDI_{MAT\_90}$) and MAT 10th percentile ($TDI_{MAT\_10}$), with largest correlations obtained between SGC and TDI derived with MAT ($TDI_{MAT}$: $R^2 = 0.52$; $TDI_{MAT\_10}$: $R^2 = 0.29$; $TDI_{MAT\_90}$: $R^2 = 0.30$, Fig. 4 and Supplementary Fig. 3). Furthermore, we did not find large differences between conditional ($R^2 = 0.57$) and marginal ($R^2 = 0.52$) effects after accounting for the taxonomic bias in our results. Overall, these results indicate that the displacement from species $T_{opt}$ is a good predictor of the magnitude of the observed species level growth responses to temperature change at our experimental sites. The relationships between scaled growth rate (SGR) and any metric of TDI among species groups are opposite in direction and there is a significant difference between the absolute value of the slopes of these relationships ($-0.08 \pm 0.01$ for montane and $0.07 \pm 0.02$ for lowland species for the case of TDI estimated with MAT, $p = 0.0001$, Fig. 4) being steeper for montane species.

## Discussion

Our findings demonstrate that temperature alone does not explain the overall reduction in montane species presence at the hotter extremes of their thermal distributions, as reported in observational distribution studies[14,15,39]. Although 55% of montane trees did not survive at the extreme of their thermal ranges, 45% (from nine out of the eleven planted dominant montane species) did survive under these conditions: four species decreased growth and five species did not change growth when growing at their hot extreme, which is on average 6.5 °C higher than their $T_{opt}$. This indicates that some dominant montane species can tolerate a certain amount of increased warming. These results challenge the overall assumption that tropical montane trees are universally highly sensitive to warming and are

unable to survive in hotter conditions[14,15,39,40]. A separate dataset from this experiment[41] demonstrated that some, but not all, montane species can acclimate their photosynthetic capacity and leaf respiration to warming when growing under the hot extreme of each species thermal range. Together, these findings suggest that the assumption of universal physiological limitation of montane trees to warming is insufficient to account for observed patterns in thermophilisation along tropical elevational gradients because some montane species can acclimate[41]. However, under these conditions, i.e., at the hot extreme of montane species tolerances, lowland species grow faster than montane species, presumably competing strongly for space and resources against montane species[42].

Overall, our experimental findings highlight that rising temperatures pose a real threat to many tropical plants despite of not controlling for all possible factors that vary under natural montane forest settings. Although our experimental sites are located in areas with high precipitation (>2000 mm yr⁻¹) with relatively short rainy season (maximum ~ 21 consecutive days without rain), our study has limitations in understanding the effects of differing levels of water availability on survival and growth and the effects of precipitation variability in the Andes[4,11-13]. Under the natural settings of a tropical elevation gradient is also not possible to control for (i) indirect temperature effects via vapour pressure deficit which affects photosynthesis and water use efficiency and (ii) incident radiation due to variations in cloud cover which influences tree growth and mortality. Nevertheless, the assumption of annual temperature as the main driver of species performance alongside elevation in a high precipitation region is still valid[43]. Although soil conditions were controlled in our experiment, root expansion to local soils is expected over time, and the slightly differing soil nutrients may influence tree performance once trees reach a suitable size[44]. High topographic complexity can provide a diverse climatic space, which is crucial for species persistence. Experimental sites were selected with similar topography to reduce differing topographic effects on tree growth[45]. Furthermore, our study does not account for potential changes in biotic interactions due to warming or cooling, which could influence survival (e.g., herbivory[35]), growth (e.g., mycorrhizal associations[46]), and reproduction (e.g., pollination[47]). Since the experiment used controlled soil conditions and constant watering, the results may not fully represent the complex interactions present in natural, variable environments. Our study includes 15 of the 37 (40%) most dominant species from the Colombian Andes which belong to 8 of the 13 dominant taxa, implying that although our experiment has a high representation of dominant taxa (61%), species responses may be shared among close relatives.

In the highlands, where montane species are abundant, lowland species such as the four lowland species from the same genera used in this study can survive, albeit with reduced growth rates. This could suggest that during prior periods of warming, lowland species may have been able to colonize high-elevation areas despite experiencing temperatures below their thermal optimum[29]. However, their slow growth rates under these conditions likely prevent lowland species from outcompeting montane species. In our experiment, trees were growing in sun exposed open areas, where minimum temperatures are extreme with strong wind exposure mainly reducing lowland species performance[48] at the 14 °C MAT experimental site. However, these results do not apply to under canopy environments which are mostly shaded conditions. In forest plots, lowland species growing under the canopy are observed to be increasing growth and abundance in high elevations[14,15,20]. Therefore, under future warming it could be expected that lowland species will form part of the highland tree communities, potentially increasing local species richness due to the large number of species found in lowland forests[18]. However, the rate of species compositional shifts, the expansion of lowland species, and the spread of heat-tolerant montane species, as well as their consequences for tropical Andean communities, will likely depend on the pace of climate change[10].

The observed pattern of thermophilisation across Andean forests shows a heterogeneous trend, with rapid changes in tree composition at some sites but slower rates at others; in certain locations, montane tree abundance has even been observed to increase[14,15]. Although this

experiment indirectly evaluates interspecific competition effects, our results could provide a possible explanation for this pattern of heterogeneity: the replacement rate of montane species which die, i.e. those unable to tolerate increased warming, with lowland species, is affected by the presence of the dominant heat-tolerant montane species, which can maintain or increase their abundance under moderate warming. It suggests that the dominant heat-tolerant montane species can reduce the rate of thermophilisation where they are present, potentially providing resilience to climatic warming for these sensitive ecosystems.

Large species variability across experimentally applied temperature treatments in transplant studies has been observed, with effects ranging from positive to negative on survival, photosynthesis, growth and leaf functional traits[19,41,48,49] supporting results obtained in this study. Such species differences may be related to different growth strategies[50]. For example, large variation in sensitivity to warming was found in a transplant experiment with Afromontane forest tree species in Rwanda[19]. The survival of all studied species declined with rising MAT. However after establishment, early successional species showed faster growth with a 2 °C increase in MAT, while some late successional species exhibited either no response or reduced growth[19]. In a warming experiment involving boreal and temperate forest species, it was found that species growing close the hot extreme of their thermal ranges exhibited reductions in net photosynthesis and growth, whereas species growing closer to their cold extreme responded positively to warming[49]. Similarly, a seedling transplant experiment with dominant canopy-forming treeline species in the southern tropical Andes revealed species-specific differences and contrasting responses in seedling survival to changes in MAT[48]. Furthermore, tree growth is the outcome of photosynthesis (carbon gains) minus plant respiration (carbon losses) and both processes have been shown to moderately acclimate to warming in tropical montane species[37,51,52]. If both photosynthesis and respiration acclimate to changes in temperature, but tree growth is reduced, plants might be investing the net carbon gains in additional metabolites to thermoregulate, increase thermal tolerance or increasing root biomass to increase water uptake, but these hypotheses need to be tested. Our findings suggest considerable variation in heat sensitivity among coexisting species, linked to their thermal distribution ranges (Fig. 4, Supplementary Fig. 3) and potentially influenced by physiological temperature sensitivities.

We found that the magnitude of the observed growth responses to temperature change can be explained by species' thermal distributions which were estimated from coarse-scale air temperature records[53]. Our analysis is in agreement with other studies that have used air temperature derived from global datasets such as Worldclim from species geographical occurrences to estimate thermophilisation metrics in large-scale plant ecology studies in tropical mountains[14,15,20,54]. Furthermore, the hot extreme of a species geographical range has been suggested as a stronger physiological barrier than their cold etreme[26]. We therefore suggest that thermophilisation is an outcome of the asymmetric response of montane and lowland species to their hot and cold extremes under future warming. Since the thermal distribution obtained from geographical occurrences is representative of species growth variation, the amplitude of such thermal distributions the standard deviation of temperature across the geographical range in the denominator of TDI, Eq. (3) can be then used as a macroecological surrogate of species thermal tolerance, thus, influencing tree growth.

The future thermal tolerance of tropical montane forests depends, among other factors, on the rate of directional change of community composition (i.e., thermophilisation) and the heat tolerance of montane trees comprising the modified communities. Findings from this study advance our understanding of the mechanisms of thermophilisation as follows: (i) maintenance of heat-tolerant montane species; (ii) increase loss of heat-sensitive montane trees; (iii) strong competition of coexisting heat tolerant montane and lowland species in locations where their hot and cold ranges respectively coincide; and (iv) upslope migration of lowland species towards locations where under current climate they are uncompetitive (high elevation). Collectively, these insights suggest there is a high risk of loss of heat-sensitive montane species and that only a few heat-tolerant montane species will be able to increase their abundance under future warming. The consequence would be a forest community with a less diverse, more homogenous range of montane tree species. We therefore conclude that whilst tropical montane forests will be partially resilient to future warming, that there is potential for their uniquely high levels of biodiversity to be substantially compromised, and at large scale their ecosystem functioning may be jeopardized by community homogenization[55,56].

## Methods

### Species thermal distributions

This study focuses on dominant mid-successional tree species from the Colombian Andes, characterised by intermediate wood density values (0.46–0.6 g cm$^{-3}$) and sun-germination capability. Species were filtered from a large dataset of 115 forest plots ranging in elevation from 1000 to 3750 masl from the Colombian Andes, gathered as part of the COL-TREE Network (https://coltree.com.co/) on which 40 dominant species (including three palm species) represent 50% of the total number of stems in this dataset. For each species (selected from the Colombian Andes data set), we estimated their geographical thermal range based on biological records and global interpolated climatic data from the whole of the Andes region which encompasses the whole geographic distribution area of each species. We used 6958 spatially unique observed species presence records at 30 arc seconds (~1 km) reported in the botanical information and ecology network, BIEN[57] which provides access to a number of different public datasets including the Global Biodiversity Information Facility (GBIF)[57]. To avoid multiple clustered observations, only one record per grid cell was used.

Tree records were filtered based on the difference between elevation provided in the raw GBIF data and the elevation extracted from a digital elevational model based on their geographic coordinates: If such elevation difference was larger than 200 m, the GBIF record was excluded from further analyses. From locations where species occurrences were registered, for each species occurrence, we recorded the mean annual temperature (MAT), the minimum temperature ($T_{min}$) defined as the average minimum temperature of the coldest month and the maximum temperature ($T_{max}$) defined as the average maximum temperature of the warmest month ($T_{max}$) from the WorldClim V.2. dataset[58] reported for the period 1970–2000. We then determined species level temperature distributions across the Andean region: species thermal optimum ($T_{opt}$) was estimated as the average of MAT from all available records for each species and $T_{min}$ and $T_{max}$ were represented by the variability observed in the respective records, reflecting the range of temperatures each species experiences (Fig. 1, Table 1). Variations in $T_{min}$ and $T_{max}$ delimit the cold and warm portion/range of each species thermal range respectively and are used to define the cold extreme (temperatures between the 25$^{th}$ and the 10$^{th}$ percentile of $T_{min}$) and the hot extreme (temperatures between the 75$^{th}$ and the 90$^{th}$ percentile of $T_{max}$) of the thermal distribution of each species. We removed 5% of data in each extreme of the distribution (5% and 95%) to minimise estimation bias in $T_{opt}$ and the cold and hot extremes.

### Montane and lowland species

Andean species are distributed within a wide elevation/temperature range, therefore different responses to changes in temperature are expected[14]. Two groups of species were observed on the estimated species' thermal distributions with varying values of $T_{opt}$ associated to low and high thermal environments. For this reason, we performed a cluster analysis (k-means) across temperature and elevation in the whole tropical Andes from 500 to 3500 masl to partition species' thermal space to identify a breakpoint in terms of temperature and elevation between them. We divided thermal space across the whole tropical Andes into two groups: a low temperature group, with mean temperature of 13.7 (±2.7) °C and altitude of 2545 (±457) masl, and a high temperature group with mean temperature of 22.02 (±2.2) °C and altitude of 1108 (±396) masl. The breakpoint between these two groups was found at 18 °C and 1825 masl (Supplementary Fig. 4) Therefore, in this study, we define montane tree species as those with $T_{opt}$ lower than

**Table 1 | Study species, their thermal and elevation distributions and wood density**

| Species | GBIF record | $T_{opt}$ (°C) | *Mean $T_{min}$* (°C) | *Mean $T_{max}$* (°C) | Thermal range (°C) | $T_{SD}$ (°C) | Cold extreme: 10th and 25th % tiles of $T_{min}$ (°C) | Hot extreme: 75th and 90th % tiles of $T_{max}$ (°C) | Min Altitude (masl) | Max Altitude (masl) | Wood density (g.cm⁻³) |
|---|---|---|---|---|---|---|---|---|---|---|---|
| *Inga spectabilis* | *314* | 25.1 ± 2.24 | 19.5 ± 2.5 | 30.1 ± 2.4 | 11.1 | 2.2 | 15.9–17.8 | 32.3–34.2 | 0 | 1500 | 0.58 |
| *Inga ingoides* | *295* | 24.7 ± 2.6 | 17.7 ± 3.5 | 31.1 ± 2.6 | 14.9 | 2.6 | 13.9–15.7 | 32.7–33.3 | 0 | 2000 | 0.49 |
| *Inga marginata* | *856* | 22.9 ± 3.3 | 15.3 ± 5.1 | 30.3 ± 2.7 | 15.2 | 3.4 | 8.2–10.1 | 31.9–34.7 | 0 | 2000 | 0.58 |
| *Inga densiflora* | *302* | 22.4 ± 3.3 | 16.8 ± 3.2 | 28.3 ± 3.6 | 11.8 | 3.3 | 12.3–14.6 | 31.2–32.3 | 500 | 2500 | 0.58 |
| Miconia theizans | 733 | 16.9 ± 3.7 | 10.3 ± 4.1 | 23.1 ± 4.1 | 14.1 | 3.7 | 5.3–7.1 | 26.5–28.4 | 500 | 3000 | 0.62 |
| Ilex laurina | 123 | 16.7 ± 3.2 | 11.3 ± 3.2 | 22.1 ± 3.6 | 10.5 | 3.2 | 7.3–9.9 | 24.3–26.2 | 1500 | 2500 | 0.55 |
| Guatteria lehmannii | 22 | 16.5 ± 1.4 | 11.8 ± 1.2 | 21.3 ± 1.6 | 10.1 | 1.4 | 10.4–10.9 | 22.2–23.3 | 1500 | 2500 | 0.56 |
| Hieronyma antioquensis | 75 | 16.4 ± 2.9 | 11.7 ± 2.9 | 21.2 ± 3.1 | 9.0 | 3.0 | 9.9–10.3 | 21.3–24.0 | 2000 | 2500 | 0.63 |
| Andesanthus lepidotus | 1307 | 15.7 ± 3.1 | 10.6 ± 3.3 | 20.9 ± 3.2 | 11.9 | 3.1 | 6.26–7.8 | 22.7–25.0 | 1000 | 3500 | 0.63 |
| Clethra fagifolia | 470 | 15.6 ± 2.3 | 10.6 ± 2.4 | 20.8 ± 2.5 | 10.4 | 2.3 | 7.3–10.0 | 21.8–23.9 | 500 | 3500 | 0.48 |
| Chrysochlamys colombiana | 88 | 15.6 ± 2 | 10.6 ± 2 | 20.6 ± 2.2 | 10.0 | 2.0 | 8.4–9.3 | 22.3–22.9 | 500 | 3000 | 0.43 |
| Quercus humboldtii | 701 | 15.4 ± 1.9 | 11.0 ± 2.7 | 20.1 ± 1.5 | 9.4 | 3.3 | 7.4–11.0 | 21.2–22.5 | 1500 | 3500 | 0.69 |
| Weinmannia pubescens | 521 | 14.8 ± 1.9 | 9.9 ± 1.9 | 19.7 ± 2.3 | 10.6 | 1.9 | 7.5–9.4 | 20.6–2.6 | 1500 | 3000 | 0.50 |
| Clusia multiflora | 1040 | 14.7 ± 3.1 | 9.6 ± 3.3 | 19.8 ± 3.3 | 12.0 | 3.1 | 5.0–7.1 | 20.8–24.1 | 1000 | 3500 | 0.56 |
| Clusia ducu | 111 | 13.5 ± 1.5 | 7.3 ± 1.6 | 18.7 ± 1.7 | 12.0 | 2.2 | 6.2–6.6 | 19.0–20.9 | 1500 | 3000 | 0.56 |

GBIF record: number of records from GBIF used to calculate each species thermal and elevational range, species thermal optimum, $T_{opt}$, is estimated as the species level average value of mean annual temperature (MAT) from all records, mean $T_{min}$ and $T_{max}$ correspond to MAT of the coldest and warmest month respectively from the WorldClim V.2. dataset[58] for the period 1970–2000, with corresponding thermal range (mean $T_{max}$- mean $T_{min}$), standard deviation of species thermal range ($T_{SD}$), hot and cold extremes (Hot extreme, Cold extreme) of each species thermal distribution, minimum (Min Alt) and maximum (Max Alt) altitude. Lowland species are in italic and Montane species are in roman.

18 °C which have a thermal distribution range between 6 °C and 24 °C, and lowland tree species as those with $T_{opt}$ higher than 18 °C which have thermal distribution range between 15 °C and 32 °C (Table 1).

**Study species**

We selected 15 dominant tree species from the Colombian Andes of intermediate succession (out of a total of 37) that fulfilled the following criteria: species with (i) both wide and narrow ranges of observed thermal distributions, estimated as the differences between species level average $T_{max}$ and $T_{min}$; (ii) both high and low $T_{opt}$ in those distributions; (iii) species that belong to the most abundant genera in the Neotropics[59] (e.g., *Inga*, *Guatteria*, *Miconia*, *Clusia* and *Weinmannia*). Based on the above criteria, we selected a total of four lowland and eleven montane species with $T_{opt}$ between 13 °C and 25 °C with thermal ranges (difference between average $T_{max}$ and $T_{min}$) between 9 °C and 15 °C. The lowland group includes two species with $T_{opt}$ close to 22 °C, and two species with $T_{opt}$ close to 26 °C. The montane group consists of eleven species whose $T_{opt}$ is close to 14 °C with average $T_{max}$ up to 22 °C, except one species which average $T_{max}$ goes up to 24 °C (Fig. 1, Table 1).

**Study area**

The study was conducted within the *Montane-Acclim* project (https://andeantreewarming.wordpress.com/), a large-scale natural warming experiment setup to investigate dominant tropical montane forest tree species responses to warming. Experimental locations were selected based on (i) thermal ranges of the selected species, i.e. sites with mean annual temperatures that allowed hypothesis testing and on (ii) logistical constraints including access to private land with owner's approval for planting and to collect data. The project includes three experimental common garden tree plantations established along a temperature/elevational gradient in the western range of the Colombian Andes (Supplementary Fig. 1) where mean

annual precipitation is consistently high and above 2000 mm year⁻¹. Mean annual temperature (MAT) at experimental sites corresponds to 14 °C, 22 °C and 26 °C recorded by a weather station together with other variables during the period October 1ˢᵗ, 2019 to Jan 31ˢᵗ, 2022 (Table 2). Experimental sites were named after their MAT. The 14 °C experimental site (2516 masl, 2774 mm year⁻¹, latitude: 5.513277 N, longitude: -75.678311 W) is in the municipality of Supía in the Caldas province, near the San Lorenzo indigenous reserve, a regional conservation area of upper montane forest. Most species used in this study naturally grow in the neighbouring montane rainforest to this site where seeds were collected from. The 22 °C experimental site (1357 masl, 2045 mm year⁻¹, latitude: 5.641678 N; longitude: -75.685954 W) is located c. 17 km North of the 14 °C experimental site and is within the Tamesis municipality (Antioquia), near the regional protected area Cuchilla Jardin-Tamesis of lower montane forest. The 26 °C experimental site (736 masl, 2298 mm year⁻¹, latitude: 5.844561 N; longitude: -75.710442 W) is located c. 20 km North of the 22 °C experimental site and is in the proximity of the Cauca River within Puente Iglesias, Fredonia (Antioquia), corresponding to premontane forest. Sites with similar topography were selected to minimize its influence on plant performance.

**Experimental setup**

All species were exposed to MAT within or close to the thermal optimum ($T_{opt}$) of their observed natural thermal distribution, i.e., 14 °C for montane and 22 °C or 26 °C for lowland species (Table 1). Additionally, montane species were exposed to MAT within the hot extreme of their natural thermal ranges (22 °C) but also outside the hot extreme of these ranges (26 °C), the latter simulating the impact of potential future warming (Fig. 1). In contrast, lowland species (two species with $T_{opt}$ close to 22 °C and two species with $T_{opt}$ close to 26 °C) were exposed to MAT within the cold portion of their thermal range and to their cold extremes (14 °C, except for one species with a very low cold extreme not covered within $T_{min}$ variations

**Table 2 | Weather data and characteristics of native soils at experimental sites**

| Site characteristics | 14 °C site | 22 °C site | 26 °C site |
|---|---|---|---|
| Latitude<br>Longitude | 5.513°N<br>−75.678°W | 5.541°N<br>−75.685 W | 6.844°N<br>−75.810°W |
| Elevation (masl) | 2516 | 1357 | 736 |
| MAT (°C) | 13.78 | 22.14 | 25.58 |
| $T_{day}$ (°C) | 16.1 | 22.5 | 27.1 |
| $T_{night}$ (°C) | 12.6 | 19.6 | 22.7 |
| MAT (10%; °C) | 11.6 | 18.3 | 20.7 |
| MAT (90%; °C) | 18.4 | 26.4 | 32.3 |
| MAP (mm yr$^{-1}$) | 2774 | 2045 | 2298 |
| VPD$_{day}$ (kPa) | 0.82 | 1.14 | 1.83 |
| VPD (90%; kPa) | 1.57 | 2.24 | 3.17 |
| Direct PAR mean (5% — 95%; μmol m$^{-2}$ s$^{-1}$) | 580 (17–1507) | 682 (4–1864) | NA |
| Diffuse PAR mean (5% — 95%; μmol m$^{-2}$ s$^{-1}$) | 368 (20–809) | 332 (3–768) | NA |
| Maximum number of consecutive days without rain | 12.5 | 21.4 | 20.3 |
| Soil properties | | | |
| Phosphorus P (Kgha$^{-1}$) | 10.5 | 29.15 | 13.95 |
| Nitrogen N (Kgha$^{-1}$) | 231 | 204.5 | 201 |
| pH | 5.1 | 5.3 | 5.3 |

Weather data include mean values for the period October 1$^{st}$, 2019, until January 31$^{st}$ 2022. Mean daytime ($T_{day}$) and night-time ($T_{night}$) temperatures were calculated from 06:00–17:59 and 18:00–05:59 respectively. Mean annual temperature (MAT) and mean annual precipitation (MAP) include the 1$^{st}$ and the 99$^{th}$ percentiles. Mean daytime vapour pressure deficit (VPD$_{day}$) was calculated using 06:00–17:59 values. Native soil data at experimental sites represent the mean of the top 0–30 cm taken from three random samples at each experimental site.

at the 14 °C site), simulating their potential upward migration as suggested in thermophilisation studies[39]. Lowland species were also exposed to temperatures within the hot portion of their thermal range including the hot extremes, covered by the large temperature variation at the warmest experimental site (26 °C).

At each field site, trees were arranged in four 600 m$^2$ plots with six 94 m$^2$ blocks within each plot. Each block contains one tree of each of the 15-study species, planted 2.5 m apart from each other to avoid competition during the first three to four years of growth at experimental locations. This adds up to six trees of each species in each plot, for a total of 24 trees of each of the 15 species per site, with a total of 360 trees at each site. The position of each tree in each block was randomised across species. In addition, one of the six blocks was used as a nutrient availability treatment. Information on fertiliser addition and timings is presented in Supplementary Notes 1 and Supplementary Tables 2-5 Seeds of all species were collected from the forest neighbouring the 14 °C experimental site. Specifically, seeds from montane species were collected from elevations ranging between 2200 and 2500 masl, within temperatures close to their $T_{opt}$, while lowland species were collected from elevations between 1300 and 2200 masl, within temperatures close to the cold portion of their geographic range. All seeds were collected from a minimum of three to a maximum of five trees per species to minimise intra-specific variation. All seeds were propagated in poly-pots in a nursery located at a site with mean annual temperature of 22 °C and a minimum of 100 seedlings per species were produced. Trees were planted during November 2018 in open areas at each experimental site following 8–24 months of growth in a nursery. Planting height varied between 50 and 100 cm (depending on species). Trees were planted in 0.32 m$^3$ (0.8 × 0.8 × 0.5 m) soil pits using soil extracted from a nearby location to the 14 °C site (400 kg of soil were used per tree), to maintain soil physical and chemical

conditions (Supplementary Tables 6-7). All trees were tagged, mapped, and their diameter at 2 cm height was marked to ensure all posterior measurements were taken at the same stem position. If there were any stem irregularities, the 2 cm mark was moved to a nearby location. All trees have been irrigated since planting, using an average of 8–24 l of water per night when there were no rain events during two consecutive days to allow successful plant establishment and avoid effects of water limitation. See Table 2 for maximum number of consecutive days without rain at experimental sites.

## Growth and survival monitoring

Trees were monitored from February 2019 until January 2022 and their survival and diameter were recorded. Measurements were taken every four months and data from nine measurement campaigns were used in this study[60]. For survival, each individual tree was classified as alive if its stem was green even if it did not have any leaves. Tree diameter ($D$) was used to calculate the relative growth rate (RGR – a metric that indicates the proportion of growth per unit of time) as the difference between the logarithm of the tree diameter at $i$ census ($D_i$) and the diameter taken during the first census ($D_0$) divided by the time interval between measurements ($t_i – t_0$): The growth rate per tree was derived as follows[52]:

$$RGR = [\log(D_i) - \log(D_0)]/(t_i - t_0) \qquad (1)$$

With $D$ expressed in millimetres (mm), ($t_i – t_0$) in years and RGR in mm mm$^{-1}$ year$^{-1}$.

To allow comparisons across species, we scaled the RGR for each species dividing by the maximum RGR reported for each species in all sites, hereafter referred as the scaled growth rate (SGR) at species level[52]:

$$SGR = RGR_{ij}/max(RGR_j) \qquad (2)$$

Where SGR is the scaled growth rate at species level of individual tree $i$ from species $j$, RGR is the relative growth rate from an individual tree $i$ from species $j$ and $max$ (RGR$_j$) is the maximum reported relative growth rate for species $j$. The units of both the numerator and the denominator of Eq. (2) cancel each other out and SGR at species level is unitless and values range between zero and one.

We evaluated the relationship between scaled growth rate at species level (SGR, Eq. (2)) and thermal displacement index (TDI)[53] for each species at each experimental site. We estimate TDI based on three MAT metrics (MAT metric) of experimental site temperature (MAT, MAT 90$^{th}$ percentile and MAT 10$^{th}$ percentile) and each species $T_{opt}$ and the standard deviation of species geographic thermal range ($T_{SD}$, Table 1) as follows:

$$TDI = (MAT\ metric - T_{opt})/T_{SD} \qquad (3)$$

## Statistical analyses

We used Cox Proportional Hazard (CPH) regression to model species survival over time for each species and experimental site. The CPH regression is a semi-parametric model that allows the quantification of predictors on the rate of event incidence (e.g., death) at a particular point in time. The CPH regression is expressed by the hazard function or force of mortality and can be interpreted as the risk that mortality events occur. In this case, it calculates the probability that an individual trees dies at a particular time at each site[37,38]. We analysed survival probability and RGR for each species across experimental sites MAT. Each experimental site represents a thermal treatment and was included as covariate in all statistical models used in this analysis representing temperature. All statistical analyses were done in R version 4.2.1.

In addition, we used two-way ANOVA to test growth rate variation across species and sites in 489 trees. Nutrient fertilisation was also included as an additional factor in the ANOVA analysis to assess the interaction of soil conditions with temperature and growth. A multi-comparison Tukey test was performed to look for significant differences across species, sites,

and fertilisation treatment. Fertilisation did not have any significant effects on tree growth or survival, indicating that the direct effect of temperature was important, and that observed effects were not mediated through indirect temperature effects on soil nutrient availability. We employed a Welch ANOVA to account for differences in sample sizes between fertilised and non-fertilised trees and found there is no impact of sample size ($F = 3.24$, $p = 0.21$). Also, we compared the growth rate of montane and lowland species at the 22 °C site using a $t$ test.

To evaluate the relationship between growth rate (SGR) and the thermal displacement index (TDI), where each observation represents one species per site, we employed a mixed-effect linear model. We also added species group as a covariate (fixed factor) to assess for differences in species response. We run a model for TDI with each of the three MAT metrics (MAT, $MAT^{90th}$ and $MAT^{10th}$ percentiles). Additionally, we used the botanical family as a random factor to control for taxonomic bias in our sampling, where all lowland species belong to genus Inga (Fabaceae), and three montane species belong to the Clusiaceae family. We reported conditional (fixed plus random effects) and marginal (only fixed effect) $R^2$ and compare both to detect the effect of taxonomic bias in our results.

## Data availability

Tree datasets measured as part of this study are available at https://doi.org/10.5285/c7ce1610-aba3-4a09-bf7c-1b6c774d597a.

## Code availability

R code (version 4.2.1) for data analysis and figure production is available at https://zenodo.org/records/11086569.

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

## Acknowledgements

Z.R. was supported by a studentship from the University of Antioquia and by the UK Natural Environment Research Council (NERC) awards NE/R001928/1 and NE/X001172/1. L.M.M., I.P.H., P.M., J.C.V., A.S., D.R.C. and S.G.C. acknowledge funding from the UK NERC awards NE/R001928/1 and NE/X001172/1. We are thankful to Juan Carlos Soto Molina and Maria Elena Botero Ospina, the Arcila- Salazar family and Cerro Prieto Colombia S.A. for providing space in their land for the experimental sites of this study. For open access, the author has applied a 'Creative Commons Attribution (CC BY) licence to any Author Accepted Manuscript version arising. The authors thank the reviewers for their valuable feedback during the review process.

## Author contributions

The study was conceived and designed by L.M.M., Z.R., I.P.H. and P.M. Z.R. collected the seeds in the forest, organised the purchase and movement of soils across experimental sites and planting of all trees across experimental sites, conducted all field measurements and analysed the data with important contributions from L.M.M., I.P.H., J.C.V. and S.G.C. and wrote the first draft of the manuscript. All co-authors contributed to the manuscript text. L.M.M., I.P.H., P.M., with support from J.C.V., A.S., D.R.C. and Z.R. wrote the research grants that funded the research.

## Competing interests

The authors declare no competing interests.
