## [Transparent Peer Review file · Communications Earth & Environment]

Heterogeneous thermal tolerance of dominant Andean montane tree species

Corresponding Author: Professor Lina Mercado

Version 0:

Decision Letter:

Dear Professor Mercado,

Your manuscript titled "Heat tolerance of tropical montane trees to future warming" has now been seen by 3 reviewers, whose comments are appended below. You will see that they find your work of some potential interest. However, they have raised quite substantial concerns that must be addressed. In light of these comments, we cannot accept the manuscript for publication, but would be interested in considering a revised version that fully addresses these serious concerns. Specifically, we ask you to:

1. Provide a more articulated context of the study in the introduction, including definitions of specific terminology and assumptions; discuss the novelty in the experimental approach considered in a balanced tone, acknowledging previous studies in the literature.
2. Ensure that the experimental setup, measurements carried out and statistical analyses are fully described and justified in the Method section; discuss limitations of the experimental setup.
3. Provide firm and robust evidence to demonstrate the changes in tree species composition, growth and mortality in tropical mountain forests in the Colombian Andes in response to global warming.

We hope you will find the reviewers' comments useful as you decide how to proceed. Should additional work allow you to address these criticisms, we would be happy to look at a substantially revised manuscript. If you choose to take up this option, please either highlight all changes in the manuscript text file, or provide a list of the changes to the manuscript with your responses to the reviewers.

When resubmitting, please provide a point-by-point response to the reviewers' comments. Please submit your responses as a separate file, distinct from your cover letter where you can add responses to the Editors' comments that you do not want to be made available to the reviewers. Word files are preferred.

Important: The response to reviewers must not include any figures, tables or graphs. If you wish to respond to the reviewer reports with additional data in one of these formats, please add them to the main article or Supplementary Information, and refer to them in the rebuttal. Due to current technical limitations, any figures, tables, or graphs embedded in your rebuttal will not be included in the peer review file, if published.

If the revision process takes significantly longer than three months, we will be happy to reconsider your paper at a later date, as long as nothing similar has been accepted for publication at Communications Earth & Environment or published elsewhere in the meantime.

Please use the following link to submit your revised manuscript, point-by-point response to the reviewers' comments with a list of your changes to the manuscript text (which should be in a separate document to any cover letter), a tracked-changes version of the manuscript (as a PDF file) and any completed checklist:

Link Redacted

Please do not hesitate to contact us if you have any questions or would like to discuss the required revisions further. Thank you for the opportunity to review your work.

Best regards,

Rossella Guerrieri, PhD
Editorial Board Member
Communications Earth & Environment
orcid.org/0000-0001-5247-0432

Alice Drinkwater, PhD
Associate Editor
Communications Earth & Environment

EDITORIAL POLICIES AND FORMAT

If you decide to resubmit your paper, please ensure that your manuscript complies with our editorial policies and complete and upload the checklist below as a Related Manuscript file type with the revised article:

Editorial Policy Policy requirements
(Download the link to your computer as a PDF.)

- Behavioural and social science
- Ecological, evolutionary & environmental sciences
- Life sciences

<https://www.nature.com/documents/nr-reporting-summary.zip>

For your information, you can find some guidance regarding format requirements summarized on the following checklist: (<https://www.nature.com/documents/commsj-phys-style-formatting-checklist-article.pdf>) and formatting guide (<https://www.nature.com/documents/commsj-phys-style-formatting-guide-accept.pdf>).

REVIEWER COMMENTS:

Reviewer #1 (Remarks to the Author):

The authors seek to understand the sensitivity of tropical montane trees to climate and climate warming through a demographic study of species planted in common gardens along an elevational gradient. The elevational gradient serves as a proxy for climate warming, under the assumption that other factors that vary with elevation (rainfall, soils, biotic pressure) are not driving species survival. Indeed, steps were taken to even out some of these other factors. A unique aspect of the study was the intent to separate warming effects from competition effects on tree growth and survival, since these are parts of the species' fundamental and realized climatic niches.

The Intro section is brief and minimal in terms of the background for the study. The section does not explain certain assumptions, and involves some disconnected ideas. For instance, tree mortality is what leads to species compositional change but the text purports this backwards: "compositional changes in the tropical Andes ...ha[s]ve show[ed]n that: i) montane tree mortality has increased ...". Species composition is conflated to mortality which is incorrect.

The authors could greatly improve their explanations and underlying assumptions in the Intro section. It seems as if they presume that with warming, plants from warmer habitats will be favoured or shift their overall distribution to present-day cooler habitats, thus maintaining equilibrium with their optimum growing temperature, and not showing physiological

acclimation to warming. If this chain of thinking underpins the work, can't this be explained in the Intro section? It is stated that 'all species were exposed to temperatures ... close to the T_{opt} of their observed natural thermal distribution'. The authors need to unfold how they came to quantify this T_{opt} , because that is crucial to the inferences coming out of this study. Very early on in the Intro section they also bring up the idea of species 'outside of their thermal ranges' without explaining how this was determined. The same applies to the statement about 'their coldest portion of their thermal range'. All of these central concepts for the manuscript need to be clarified and quantified upon first raising the ideas, at least in brief.

The second paragraph of the Intro is already digging into their specific study, but involves elements of Methods and aspects in the Methods that are raised prematurely (such as the thermal distribution means, which are assumed and then explained later). These aspects of a disorganised presentation of the info in the manuscript need to be revised with appropriate clarifications. The authors' conception of 'montane specialists' is mentioned but not explained or quantified, even though this becomes central to later interpretations. The 'elevation/temperature gradient in the western Andes of Colombia with average temperatures of 14, 22, 26°C' seems to have been a valuable platform for the study, but the authors don't quantify how they came up with these temperatures and what meteorological monitoring was behind these mean temperatures (same for the temperature bands for the sites shown in Fig. 1). Quite a lot of the work is predicated on mean annual temperatures without considering peak temperatures and extremes, even though the extremes have a disproportionate role in tree mortality (Allen, Breshears and McDowell 2015, Menezes-Silva et al. 2019). Can the authors consider some other measures of temperature in their analyses as possible drivers of mortality and distribution for these species? Why is their treatment simplified to mean annual temperature (which is often stated in the manuscript as just 'temperature' or 'site temperature'; please clarify that it is MAT in all Figure captions)? Further, simply because the analysis worked out using MAT is not exclusive support for the idea that 'air temperature from species geographical occurrences can be used as a good proxy for analysis of the effect of temperature'. This statement can be easily debunked and the authors should consider backing away from the idea, as it was purely a convenience in their manuscript.

Whilst the Intro highlights that the authors will separate warming effects from competition effects on tree growth, in the end they do not separate these at all. For instance, no neighbour analysis was done to substantiate or test competition. The idea of competition is only indirectly referred in the Discussion. The authors should return to this idea in the Results & Discussion, or remove it from the text.

Detailed comments

The 'specie's thermal optima' is incorrect English spelling.

'avoid T_{opt} estimation bias': is this avoid or minimise? One can't entirely avoid the problem.

The sites are about 40 km from one another, how do soils vary amongst them, and did other notable factors vary?

T_{opt} is defined as 'the average temperature from the species' geographic thermal range'. Is this an arithmetic mean that is used to define this variable? How do you account for biases from multiple clustered observations?

Results first sentence should say ...'closest to their presumed T_{opt} ' or 'computed T_{opt} '.

The sentence 'Our results provide an explanation for this pattern of heterogeneity ...' is a run on and should be rewritten.

Reviewer #2 (Remarks to the Author):

Review of: Restrepo ... Mercado, Heat tolerance of tropical montane trees to future warming

In this manuscript the authors studied the effect of warming on growth and mortality of 15 native dominant tree species (including montane and lowland species) across an elevation gradient. The results revealed that 55% of montane species can tolerate high temperatures, while 45% cannot, with lowland species thriving in warmer conditions. They suggest that warming is leading to increased community homogeneity and thus reduced biodiversity in the biodiversity hotspot of the Andes.

The study is timely as it improves our understanding of warming responses of species from different elevation origins and how this may influence the community composition in montane areas. The study is well conducted, have an overall good design and the manuscript is well written. However, there are a number of flaws and missing information that will hinder full understanding and interpretation of the results in the current version of the manuscript. I therefore recommend revision before the paper can be accepted.

The issues are:

The species selection is well justified based on their thermal preferences. However, the authors have overlooked the potential intra-specific variations in thermal tolerance. In general, provenances from different altitudes and latitudes may exhibit distinct thermal ranges. For instance, the montane species' germplasm was gathered from elevations ranging between 1900-2500 m a.s.l. with potentially 3-4°C temperature range. These differences can have influenced the thermal ranges of the collected species. Therefore, I recommend including details about the specific altitudes from which each species was collected and discussing potential effects of provenance on the results.

One notable advantage of this experiment is the considerable effort dedicated to using uniform planting soil across all sites. Soil from a 0.32 m² pit at each tree was replaced with soil from the upper site and distributed to the other sites. However, a question arises regarding how long the roots will remain within the replaced soil space. It is plausible that a transition period within the experimental three years will occur during which the root system gradually acclimates to the original soil conditions at the various sites.

The soil quality to which the trees are adapted could also influence their performance in the soil from the upper sites, raising another question about potential differences in soil quality between sites, which preferably should be reported in the supplementary. The application of additional fertilizers in one of the six blocks is a significant assessment of soil factors. However, especially as the authors did not detect a significant treatment effect, clarification is needed on the criteria used for selecting the quantity and quality of fertilizers, as well as the sensitivity of the significance test employed.

Tree growth was assessed by measuring stem diameter at 2 cm height, from which relative growth rates (RGR) were calculated and standardized within species by dividing each individual's RGR by the maximum RGR observed within that species. However, several key details that are crucial for understanding and interpreting the results are missing. For example, the frequency of diameter measurements and how stem irregularities at such a low height (2 cm) were addressed are not specified. Moreover, RGR typically decreases with size due to self-shading, canopy closure, and other factors. Therefore, I recommend including size data in the supplementary materials, along with growth rate trends over time if available. Since mortality rates may also vary with tree size, there is an additional reason to provide data on the size distribution of the trees and analyse the data in relation to potential differences in size.

Other factors that could have influenced both mortality and growth include diurnal and seasonal variations in climate, such as day and night temperatures, VPD, and radiation. Despite regular irrigation, precipitation may also have had an impact, but specific data on this is currently unavailable. Furthermore, it remains unclear whether the temperature ranges shown in Figure 1 encompass both day and night temperatures, or solely daytime temperatures. It is surprising that more detailed climate data are missing.

The manuscript is lacking in citations of other relevant studies, particularly those involving common garden and growth experiments assessing temperature effects. While it does reference broader ecosystem-level studies, it misses comparisons with species-specific research. Given that this study connects species-level findings to ecosystem dynamics, its results should also be discussed in relation to such studies.

A few other more specific comments:

Abstract/Last line: This sentence seems partly redundant: Maybe revise to "increased homogeneity in community composition, resulting in reduced biodiversity".

Page 9 and elsewhere: A more handy and commonly used unit for RGR would be % year⁻¹.

Page 12, line 6: Text refer to Fig. 5 which do not exist. Should be Fig 4?

Page 12/13: The authors estimate that 27% of the montane species may be lost by 2050. Except for the assumed increase of 3 °C in the Andes by 2050, what is this estimate based on and how was it estimated?

Page 13 and the bottom: The estimated relatively higher sensitivity of 14% of montane species compared to lowland species. How robust is this estimated difference? Although it is clear that the direction is different the magnitude of the response is not evident.

Reviewer #3 (Remarks to the Author):

In this manuscript the authors describe the results of a warming experiment they conducted for analyzing heat tolerance of tropical montane trees along an elevation gradient in the Colombian Andes under the current context of climate warming. The results show that lowland species have higher survival rates than montane species at higher temperatures and growth rates are higher for lowland trees at higher temperatures but higher for montane trees at lower temperatures. The authors also found results vary across species. In general, these results provide valuable insights into the species composition change occurring in tropical mountains as a response to global warming. I believe that environmental scientist could find this paper relevant. In particular, the experiment conducted is very valuable since it required a lot of work and effort from the research team and provides interesting results. There are, however, some points I would like to raise regarding the

limitations of the experiment that are important to discuss. First, water is a fundamental component of how species survive and growth under a changing climate and needs more discussion. Although it is clear that water limitation was controlled for in the experiment, it is unclear whether sites received different amounts of water once precipitation is considered and if this could have had an effect on growth and survival. Ideally, the experiment could have included a control treatment and a water*temp treatment. Given the experiment is already complex as is and water was not considered as a treatment, I suggest that the authors add some discussion on the possible effects of climate change including water, not only temperature. Second, authors should address phylogenetic, genetics and epigenetics effects on the results of the experiment. For example, all lowland species are genus *Inga*, could this represent any bias in the results? Also, seeds were collected from the same forest close to the 14 degrees C site instead of along the elevation gradient. Does this mean that seeds for lowland species were collected closer to the coldest extreme of their geographic range? Could this have an effect on the results? Third, the authors claim their results provide insights into interspecific competition but this was not tested in the experiment (there aren't treatments for testing interspecific competition in the experiment). Authors should mention how interspecific interactions could influence their results and tone down their conclusions on competition. This is important since the manuscript revolves around the concept of thermophilisation. Finally, topographic effects are not mentioned in the manuscript but are very important in mountains. High topographic complexity provides a wide climatic space and could be relevant for species persistence. Please add some discussion on this topic as it could be playing a role in the variation of thermophilisation and it is related to the claim the authors make about using coarse temperature data to study community composition change. Besides addressing these points throughout the manuscript, I suggest the authors make the following changes in specific sections and lines:

- In the introduction, authors should provide a working hypothesis and predictions.
- Lines 85 to 105 are Methods. Authors should make this paragraph shorter and move detailed info to Methods.
- The introduction should include more information about what's the expected temp change for this tropical mountain range and put into global context this specific case (what's happening in other tropical mountains?).
- Add more context to your study in the introduction. How are montane tree communities changing around the tropics?
- In Methods, the authors should explain whether the data used for estimating thermal ranges per species encompass the whole geographic distribution area of each spp or if it is only based on the presence of species within the area of study. It would be useful to add the number of records used per species to get these thermal ranges, you could summarize this information on Table S1 and add it to the main text and Figure S2.
- Seeds were collected close to an elevation which MAT is 14 degrees C. Why were seeds collected close to the 14°C and not across the study area?
- Did you also measure how much water plots received by precipitation?
- Are there any phylogenetic trends in tree growth that should be considered here? For example, all four lowland spp are *Inga*. Do *Inga* trees have faster growing rates in general?
- L.63 Define thermophilisation
- L.72 Unclear what 'these conditions' are, what conditions?
- L.72-73 In 'We further tested the performance of lowland species at high elevations' explain temp component linked to elevation. High elevation = low temperature.
- L.76 'lowland trees species' == 'lowland tree species' (remove s in trees)
- L.95 My understanding is that 26 degrees C is outside of all spp temp range (including lowland spp) not close to the T_{opt} of lowland trees
- L.98 Could you provide the approximate temperature that is expected in the future in these mountains? To contextualize 26 degrees C
- L.99 'closest' == 'close' (remove t)
- L.101 From figure 1, it doesn't look like 22 degrees C is close to the coldest end of any of the lowland tree thermal ranges. All four spp reach the blue box in Figure 1 suggesting some individuals already experience average temp close to the mean temp in the colder site
- L.124 Include years of climate normal (years used in worldclim to estimate MAT, T_{min}, and T_{max})
- L.118-130. This paragraph has several typos, please revise
- L.132 Andean spp are distributed in a wide elevation range and some spp probably also on a wide geographic range? Please clarify this (it's unclear if you're considering the whole geographic distribution area of each spp or only data of the study area).
- L.131-145 Is this the reason behind sites selection? If so, please make it explicit here.
- L.134. The k-means was computed using temp and elevation data of all records of the 40 dominant spp within the study area, correct? Or was it using all temp and elevation data space within the study area (all pixels)? Please clarify. Also explain what low and high elevation mean in this context (for example, is low between 500 and 1000 m asl?)
- L.154 'based on above criteria' == 'based on the above criteria' (add the)
- L.154 You already mention you selected 15. Change to 'we selected four lowland and 11 montane species'
- L.155 what do you mean with thermal ranges between 9 and 15 degrees C, is this only for lowland trees, are 9 and 15 average T_{opt}? According to Figure 1, the lower end of thermal ranges of some spp are lower than 9 and the higher end are way higher than 15.
- L.203-204 Only one equation is needed
- L.212 'relative growth rate from and individual' == 'relative growth rate from an individual' (remove d)
- L. 227 What predictors did you use in the regressions? Please explain further.
- L.248 nine out of 11 montane spp survived where? At the 22 degrees site? Please clarify.
- L.261 check parenthesis
- L.266 remove dash between four and lowland
- L.267 with two of the growing temp you mean two sites? Unclear, please rephrase.
- L.274 Fig 3 does not show these results, is this a t-test between lowland and montane groups? Please explain here and in

Methods

- L.275-280 You could make this paragraph shorter by removing repetitive information
- L.275-284 Are these correlations? Linear regressions? Or what type of model you used for calculating these relationships? Add to Methods (perhaps in L.216). Also, correlation does not imply that you can predict a response, it only helps explain a pattern, please rephrase here and in next paragraph.
- L.299 45% of trees or 45% of species?
- L.321 I don't think these results suggest previous colonization of higher lands in warmer periods since you didn't analyze any historical data
- L.320-326 Please add references
- L.328-330 Please add references
- Add discussion on limitations of experiment
- Figure 1 caption is unclear, please rephrase this part: "These include mean annual temperature (MAT), minimum temperature (Tmin), and maximum temperature (Tmax) estimated from the WorldClim V.2. dataset19 from locations where occurrences registered for each species in the BIEN database." Also, please explain what the box and whiskers represent and whether site's temp range was also extracted from worldclim or measured on site (for how long?).
- In panel figures, please change the position of the letters (maybe next to the spp name). It is confusing to have the letters within the main plots. Also add what the letters mean in figure captions.

Communications Earth & Environment is committed to improving transparency in authorship. As part of our efforts in this direction, we are now requesting that all authors identified as 'corresponding author' create and link their Open Researcher and Contributor Identifier (ORCID) with their account on the Manuscript Tracking System prior to acceptance. ORCID helps the scientific community achieve unambiguous attribution of all scholarly contributions. You can create and link your ORCID from the home page of the Manuscript Tracking System by clicking on 'Modify my Springer Nature account' and following the instructions in the link below. Please also inform all co-authors that they can add their ORCIDs to their accounts and that they must do so prior to acceptance.

Version 1:

Decision Letter:

Dear Professor Mercado,

Your manuscript titled "Heat tolerance of tropical montane trees to future warming" has now been seen by our reviewers, whose comments appear below. In light of their advice we are delighted to say that we are happy, in principle, to publish a suitably revised version in Communications Earth & Environment.

We therefore invite you to revise your paper one last time to address the remaining concerns of our reviewers. At the same time we ask that you edit your manuscript to comply with our format requirements and to maximise the accessibility and therefore the impact of your work.

EDITORIAL REQUESTS:

*****Please take care to match our formatting and policy requirements. We will check revised manuscript and return manuscripts that do not comply. Such requests will lead to delays. *****

SUBMISSION INFORMATION:

OPEN ACCESS:

Communications Earth & Environment is a fully open access journal. Articles are made freely accessible on publication. For further information about article processing charges, open access funding, and advice and support from Nature Research, please visit <https://www.nature.com/commsenv/open-access>

Link Redacted

Best regards,

Alice Drinkwater, PhD
Associate Editor
Communications Earth & Environment
@CommsEarth

REVIEWERS' COMMENTS:

Reviewer #1 (Remarks to the Author):

I have read this manuscript in detail and also the authors's responses to the previous round of reviews. In my opinion the authors have been diligent in their revisions and have done a fine job addressing the many comments they received on their original submission. The manuscript reads well and there are few errors at this point, so the authors are congratulated on doing such a great job with revisions.

Understanding the comprehensive nature of the authors's revisions, I hope they are open to a few small but key improvements. The authors wrote:

l. 465 'However, if trees would have been growing under the more sheltered and less cool understory in highland forests, they may be able to grow faster than in open areas, ...'

In my opinion the revisions are not much more than random speculation. I think the point is that there are aspects of the open-field environment where the study was done that differ from under-canopy situations, so the results cannot apply well to these shaded conditions. I would want the authors to better reflect this particular caveat, as shaded environments are not just cooler than the open areas where they conducted their study, and the authors can be more general in their comments rather than engage in such speculation.

l. 476 'Although this experiment did not directly but indirectly evaluate interspecific competition effects ...'. This is awkward wording that can be improved.

l. 489 'For example, significant variation in sensitivity to warming was found in a transplant experiment with Afromontane forest tree species in Rwanda.'

A specific study is mentioned here, so I would expect to see a reference associated with the statement. Maybe it is ref 56? As the authors are specifically point to this I think the reference should be attached.

Thanks to the authors for producing an interesting and valuable study.

Reviewer #2 (Remarks to the Author):

The manuscript examines the effects of warming on the growth and mortality of 15 native tree species across an elevation gradient in the Andes. Results show that 55% of montane species struggle with high temperatures, while lowland species

thrive, suggesting warming drives community homogenization and reduced biodiversity in the Andes.

The study provides valuable insights into species' responses to warming and its impact on montane community composition. It clearly enhances our understanding of the potential effects of climate change on these forest ecosystems, making it highly relevant to both the scientific community and efforts in restoration and conservation.

The revised manuscript is well-structured, clearly written, and thoughtfully improved. The reviewers' comments on the initial version have been effectively addressed with well-reasoned explanations. I recommend the revised version for publication.

Reviewer #3 (Remarks to the Author):

Great job replying to all comments, congratulations on your work. No further revisions are needed.

Response to Reviewer #1 (Remarks to the Author):

We thank the Reviewer for their very useful feedback. We have numbered each of the comments and provided a response in grey text following each comment. Responses refer to the line numbers of the manuscript file with highlighted changes. We have also copied the respective manuscript text in the response below in most cases (when it was suitable). The highlighted text in grey corresponds to new text while non highlighted is text from the originally submitted manuscript.

The authors seek to understand the sensitivity of tropical montane trees to climate and climate warming through a demographic study of species planted in common gardens along an elevational gradient. The elevational gradient serves as a proxy for climate warming, under the assumption that other factors that vary with elevation (rainfall, soils, biotic pressure) are not driving species survival. Indeed, steps were taken to even out some of these other factors. A unique aspect of the study was the intent to separate warming effects from competition effects on tree growth and survival, since these are parts of the species' fundamental and realized climatic niches.

1. The Intro section is brief and minimal in terms of the background for the study. The section does not explain certain assumptions and involves some disconnected ideas. For instance, tree mortality is what leads to species compositional change but the text purports this backwards: "compositional changes in the tropical Andes ...ha[s]ve show[ed]n that: i) montane tree mortality has increased ...". Species composition is conflated to mortality which is incorrect.

Response: We have substantially extended the introduction to provide more background for the study in response to each specific comment from all reviewers. In response to the specific query on species compositional changes and mortality we modified the text to clarify this on the second paragraph of the introduction. Observed species composition changes are explained in the literature as the result of both increased mortality of montane species at lower elevations and increased abundance of lowland species across elevations via upward migrations. Lines 87-102 include a clarification of the text on changes species composition (thermophilisation) and causes as follows:

The upward movement of species in montane environments from the warm lowlands to cooler uplands produces a reconfiguration of species communities which has been termed thermophilisation¹⁵: warm affiliated lowland foothill thermophilic species, hereafter termed lowland species, are increasing in abundance across elevations relative to highland cold affiliated montane species, hereafter termed montane species. Observed directional shifts in species composition over time detected on forest plots^{14,15,19,20} provide evidence of thermophilisation on tropical montane tree communities in the Andes (reported in Colombia, Peru, Ecuador and Argentina^{14,15}), in Afromontane forests (reported in Rwanda, Uganda, Democratic Republic of Congo and Tanzania²⁰) and in Central America (reported in Costa Rica²¹ and Jamaica²²). Thermophilisation in Andean forest is consistent with concurrent warming in the region and is caused by 1) increased abundance of lowland species in their upper limit of elevational range which coincides with the cold extreme of the thermal range, expanding their elevational range^{14,15,20} and 2) increased mortality of montane species in their lower

limit of their elevational ranges (i.e. the range of elevations within which species can survive) which coincide with the hot extremes of their thermal ranges, leading to contractions of their elevational range¹⁹.

2. The authors could greatly improve their explanations and underlying assumptions in the Intro section. It seems as if they presume that with warming, plants from warmer habitats will be favoured or shift their overall distribution to present-day cooler habitats, thus maintaining equilibrium with their optimum growing temperature, and not showing physiological acclimation to warming. If this chain of thinking underpins the work, can't this be explained in the Intro section?

Response: We appreciate this perspective on the assumptions underlying our study. We do not show data on physiological acclimation in this manuscript; therefore, we refrain from posing hypotheses or assumptions on thermal acclimation of the various key processes involved (photosynthesis, respiration and stomatal conductance) since we cannot directly test thermal acclimation hypotheses with growth data. We assess thermal acclimation of physiological processes in *Cox et al (2023), New Phytol, 238(6):2329*, and on a recently submitted manuscript. Instead, in this manuscript we investigate tree growth responses (indeed affected by photosynthesis and respiration) from species from cool and warm habitats to temperature change mimicking thermophilisation. In lines 111-122 we elaborate on expectations of plant performance under extreme cooling and warming as follows:

According to Shelford's law²³, plant performance is limited by any deficit or excess of environmental conditions or resources (e.g., temperature), leading to a gradual reduction in performance from optimal environment conditions at which a plant grows best to extreme conditions under which a plant performs poorly^{24,25}. However, responses of plant performance to extreme cold and to extreme high temperatures may differ due to contrasting metabolic constraints (e.g., chilling vs. heating). The lower limit of a species' thermal distribution is considered a strong limiting factor to plant performance because cool conditions reduce metabolic rates²⁶, and thus growth. Conversely, the upper range and hot extreme of a species' thermal distribution may abruptly restrict enzymatic activity^{27,28}. Therefore, the displacement of a species from its thermal optimum towards the species' hot extreme could thus lead to a steeper reduction in tree growth compared to species displacement towards their cold extreme.

In the discussion we acknowledge our understanding of the indirect impact of thermal acclimation of photosynthesis and respiration on montane forest tree growth in lines 501-507 as follows:

Furthermore, tree growth is the outcome of photosynthesis (carbon gains) minus plant respiration (carbon losses) and both processes have been shown to moderately acclimate to warming in tropical montane species^{42,44,57}. If both photosynthesis and respiration acclimate to changes in temperature, but tree growth is reduced, plants might be investing the net carbon gains in additional metabolites to thermoregulate, increase thermal tolerance or increasing root biomass to increase water uptake, but these hypotheses need to be tested.

3. It is stated that 'all species were exposed to temperatures ... close to the T_{opt} of their observed natural thermal distribution'. The authors need to unfold how they came to quantify

this T_{opt} , because that is crucial to the inferences coming out of this study. Very early on in the Intro section they also bring up the idea of species 'outside of their thermal ranges' without explaining how this was determined. The same applies to the statement about 'their coldest portion of their thermal range'. All of these central concepts for the manuscript need to be clarified and quantified upon first raising the ideas, at least in brief.

Response: We have now introduced species thermal distribution, thermal optimum, cold and warm portion or ranges and their extremes in the context of changes in species composition within the second paragraph of the introduction (Lines 79-87) as follows:

A species' thermal distribution comprises the range of temperatures at which the species is found; within this range, the temperature at which the species grows best is known as their thermal optimum, T_{opt} (Fig. 1). We define the cold and warm portions of a species thermal range as the variation in minimum and maximum temperatures experienced by the species, respectively. The hot extreme of a species thermal range can be defined as the temperature above the 75th percentile of the maximum temperature experienced by the species, and their cold extreme as the minimum temperature below the 25th percentile of the minimum temperature experienced by the species.

We then explain how we estimate species thermal distributions T_{opt} , and their cold and hot extremes in lines 122-127 as follows:

Quantifying the range of temperatures within which a species can survive (i.e. species thermal range) is complex, it can nevertheless be estimated using herbarium records of temperature variation across species geographical range²⁹. From such thermal distributions is plausible to determine the thermal optimum T_{opt} (mean of species' thermal distribution), the minimum and the maximum temperatures experienced by each species.

We also added more detail in the Methods section named 'Species thermal distributions' in lines 188-200 as follows:

From locations where species occurrences were registered, for each species occurrence, we recorded the mean annual temperature (MAT), the minimum temperature (T_{min}) defined as the average minimum temperature of the coldest month and the maximum temperature (T_{max}) defined as the average maximum temperature of the warmest month (T_{max}) from the WorldClim V.2. dataset³⁹ reported for the period 1970-2000. We then determined species level temperature distributions across the Andean region: species thermal optimum (T_{opt}) was estimated as the average of MAT from all available records for each species and T_{min} and T_{max} were represented by the variability observed in the respective records, reflecting the range of temperatures each species experiences (Fig. 1, Table 1). Variations in T_{min} and T_{max} delimit the cold and warm portion/range of each species thermal range respectively and are used to define the cold extreme (temperatures between the 25th and the 10th percentile of T_{min}) and the hot extreme (temperatures between the 75th and the 90th percentile of T_{max}) of the thermal distribution of each species.

4. The second paragraph of the Intro is already digging into their specific study but involves elements of Methods and aspects in the Methods that are raised prematurely (such as the thermal distribution means, which are assumed and then explained later). These aspects of a disorganised presentation of the info in the manuscript need to be revised with appropriate

clarifications. The authors' conception of 'montane specialists' is mentioned but not explained or quantified, even though this becomes central to later interpretations.

Response: As mentioned above we have introduced the terminology in the second paragraph of the introduction. Following this comment from the reviewer, we moved a lot of material that was in the introduction to the methods section. We removed the term montane specialist to avoid confusion and kept the terminology of montane species.

5. The 'elevation/temperature gradient in the western Andes of Colombia with average temperatures of 14, 22, 26°C' seems to have been a valuable platform for the study, but the authors don't quantify how they came up with these temperatures and what meteorological monitoring was behind these mean temperatures (same for the temperature bands for the sites shown in Fig. 1).

Response: There is a meteorological station at each experimental site, a summary of mean climate variables recorded during the period October 1st, 2019 to Jan 31st, 2022 is now included in Table 1. Mean annual temperature recorded at the three experimental sites during that period was 13.79°C, 22.14 °C and 25.58°C, therefore we named the experimental sites following their MAT: 14°C site, 22°C site and 26°C site. This is now clarified in the text in the section called study area in lines 237-242 as follows:

The project includes three experimental common garden tree plantations established along a temperature/elevational gradient in the western range of the Colombian Andes (Supplementary Fig. 1) where mean annual precipitation is consistently high and above 2000 mm year⁻¹. Mean annual temperature (MAT) at experimental sites corresponds to 14°C, 22°C and 26°C recorded by a weather station together with other variables during the period October 1st, 2019 to Jan 31st, 2022 (Table 2). Experimental sites were named after their MAT.

The legend of Fig 1 (originally only with daytime weather station air temperature data from each experimental location) now explains that the temperature bands for the experimental sites correspond to the thermal environment at each experimental site ranging from the 10th up to the 90th percentile of the observations during the period October 1st, 2019 -Jan 31st, 2022. Relevant lines are below:

Thermal distribution of study species and range of measured air temperatures at each experimental site during the period October 1st, 2019 to January 31st, 2022.

Coloured vertical polygons represent the thermal environment measured at each experimental site with MAT (represented with vertical coloured lines) of 14°C, 22 °C, and 26°C with the lower and upper thermal limits represented by the 10th and 90th percentiles of MAT respectively. Note that the 90th percentile of the 22°C site coincides with the MAT of the 26°C site.

6. Quite a lot of the work is predicated on mean annual temperatures without considering peak temperatures and extremes, even though the extremes have a disproportionate role in tree mortality (Allen, Breshears and McDowell 2015, Menezes-Silva et al. 2019). Can the authors consider some other measures of temperature in their analyses as possible drivers of mortality and distribution for these species? Why is their treatment simplified to mean annual

temperature (which is often stated in the manuscript as just 'temperature' or 'site temperature'; please clarify that it is MAT in all Figure captions)? Further, simply because the analysis worked out using MAT is not exclusive support for the idea that 'air temperature from species geographical occurrences can be used as a good proxy for analysis of the effect of temperature'. This statement can be easily debunked, and the authors should consider backing away from the idea, as it was purely a convenience in their manuscript.

Response: The analysis presented in figures 2 and 3 does not involve the use of time series of site temperatures, it only uses tree survival and tree diameter data for all species at each site where each site is considered as a temperature treatment. To clarify this, we added the following sentence in the statistical analysis part of the method section in lines 330-332 as follows:

Each experimental site represents a thermal treatment and was included as covariate in all statistical models used in this analysis representing temperature.

The variation in site temperatures presented in figure 1 includes maximum and minimum temperatures”.

Following the reviewer’s suggestion, we have now performed the analysis originally presented in figure 4 (correlation between the scaled growth rate (SGR) and thermal displacement index (TDI) only with mean annual temperature of each experimental site) now additionally using MAT 90th and MAT 10th percentiles. We added the following text in the result section in lines 397-403 as follows:

Significant relationships were found for both groups of species when estimating TDI with MAT, MAT 90th percentile and MAT 10th percentile, with largest correlations obtained between SGC and TDI derived with MAT (TDI_{MAT} : $R^2 = 0.52$; $TDI_{MAT_{10}}$: $R^2 = 0.29$; $TDI_{MAT_{90}}$: $R^2 = 0.30$, Supplementary Fig. 3). Furthermore, we did not find large differences between conditional ($R^2 = 0.57$) and marginal ($R^2 = 0.52$) effects after accounting for the taxonomic bias in our results.

The original statement: ‘This suggests that air temperature from species geographical occurrences can be used as a good proxy for analysis of the effect of temperature in large-scale plant ecology studies^{1,2,7}’ was replaced with the following lines 512-515 as follows

Our analysis is in agreement with other studies that have used air temperature derived from global datasets such as Worldclim from species geographical occurrences to estimate thermophilisation metrics in large-scale plant ecology studies in tropical mountains^{14,15,20,58}.

The growth and survival monitoring section of the methods was modified to include calculations of TDI with the 90th and 10th percentiles of MAT (see lines 318-321)

We estimate TDI based on three MAT metrics (MAT metric) of experimental site temperature (MAT, MAT^{90th} and MAT^{10th} percentiles) and each species T_{opt} and the standard deviation of species geographic thermal range (T_{SD} , Table 2) as follows:

All figure captions refer now to MAT as the average temperature at experimental sites.

We also included text in the statistical section of the methods explaining how this part of the analysis was done in lines 344-352 as follows:

To evaluate the relationship between growth rate (SGR) and the thermal displacement index (TDI), where each observation represents one species per site, we employed a mixed-effect linear model. We also added species group as a covariate (fixed factor) to assess for differences in species response. We run a model for TDI with each of the three MAT metrics (MAT, MAT^{90th} and MAT^{10th} percentiles). Additionally, we used the botanical family as a random factor to control for taxonomic bias in our sampling, where all lowland species belong to genus *Inga* (Fabaceae), and three montane species belong to the Clusiaceae family. We reported conditional (fixed plus random effects) and marginal (only fixed effect) R² and compare both to detect the effect of taxonomic bias in our results.

7. Whilst the Intro highlights that the authors will separate warming effects from competition effects on tree growth, in the end they do not separate these at all. For instance, no neighbour analysis was done to substantiate or test competition. The idea of competition is only indirectly referred in the Discussion. The authors should return to this idea in the Results & Discussion or remove it from the text.

Response: As explained in the methods, trees were planted in open areas, 2.5 m apart to avoid competition during the first 3 to 4 years of growth at experimental sites. This has been clarified in the methods section under experimental set up with the following lines 269-271:

Each block contains one tree of each of the 15-study species, planted 2.5 m apart from each other to avoid competition during the first three to four years of growth at experimental locations.

Supplementary Table 7 includes the average size of each species after three years of planting and before planting. At experimental sites with 14°C and 22°C MAT, maximum stem height was 165 cm and 310 cm respectively and maximum canopy diameters were 97 cm and 213 cm respectively. These measurements and the distance between trees (2.5m) mean that the individual crowns are distant, avoiding competition for light between trees. We also expect that roots do not extend longer than their own crown diameter (*Enquist 2022, tree Physiol., 22(15-16): 1045*). Therefore, we do not expect competition between trees at the 14°C and 22°C MAT sites during the study period. At the 26°C MAT site, only trees from lowland species have survived, with heights varying from ~300 up to 426 cm with maximum canopy diameter varying between 212 and 452 cm. We expect that at this site, there will be competition between the four lowland species. However, our question in this research is whether growth and survival of both montane and lowland species differ when growing under the warm extreme of the montane species (22°C MAT site) and under the cold portion and cold thermal extreme of lowland species (14°C MAT site).

Detailed comments

8. The 'specie's thermal optima' is incorrect English spelling.

Response: Thanks for pointing this out we have gone through the text and correct it.

9. 'avoid T_{opt} estimation bias': is this avoid or minimise? One can't entirely avoid the problem.

Response: We replaced 'avoid' with minimise in line 200

10. The sites are about 40 km from one another, how do soils vary amongst them, and did other notable factors vary?

Response: Table 1 includes a comparison of soil P, N and Ph of the native soils at each experimental site. Overall soil nutrients (N, P) and Ph have similar values across sites with soil P varying between ~225 -260 Kgha⁻¹ and soil N varying from ~200-230 Kgha⁻¹ and pH ranging from 5.1 to 5.3. Also, these soils are from similar geological substrate origins (Andisols) related to Andean uplift. (Young, K. R. et al (2007). Tropical and subtropical landscapes of the Andes. The physical geography of South America, 8, 200-216, Graham, A. (2009). The Andes: a geological overview from a biological perspective. Annals of the Missouri Botanical Garden, 96(3), 371-385).

11. T_{opt} is defined as 'the average temperature from the species' geographic thermal range'. Is this an arithmetic mean that is used to define this variable? How do you account for biases from multiple clustered observations

Response: This is now clarified in lines 188-195 as follows:

From locations where species occurrences were registered, for each species occurrence, we recorded the mean annual temperature (MAT), the minimum temperature (T_{min}) defined as the average minimum temperature of the coldest month and the maximum temperature (T_{max}) defined as the average maximum temperature of the warmest month (T_{max}) from the WorldClim V.2. dataset³⁹ reported for the period 1970-2000. We then determined species level temperature distributions across the Andean region: species thermal optimum (T_{opt}) was estimated as the average of MAT from all available records for each species

And in Line 184:

To avoid multiple clustered observations, only one record per grid cell was used

12. Results first sentence should say ...'closest to their presumed T_{opt} ' or 'computed T_{opt} '.

Response: Following this suggestion we have implemented the word **computed** T_{opt} in Line 355

13. The sentence 'Our results provide an explanation for this pattern of heterogeneity ...' is a run on and should be rewritten.

Response: We rewrote the sentence, and it now reads in lines 476-481 as follows:

Although this experiment did not directly but indirectly evaluate interspecific competition effects, our results could provide a possible explanation for this pattern of heterogeneity: the replacement rate of montane species which die, i.e. those unable to tolerate increased warming, with lowland species is affected by the presence of the dominant heat-tolerant montane species, which can maintain or increase their abundance under moderate warming.

Response to Reviewer #2

We thank the Reviewer for their very useful feedback. We have numbered each of the comments and provided a response in grey text following each comment. Responses refer to the line numbers of the manuscript file with highlighted changes. We have also copied the respective manuscript text in the response below in most cases (when it was suitable). The highlighted text in grey corresponds to new text while non highlighted is text from the originally submitted manuscript.

In this manuscript the authors studied the effect of warming on growth and mortality of 15 native dominant tree species (including montane and lowland species) across an elevation gradient. The results revealed that 55% of montane species can tolerate high temperatures, while 45% cannot, with lowland species thriving in warmer conditions. They suggest that warming is leading to increased community homogeneity and thus reduced biodiversity in the biodiversity hotspot of the Andes.

The study is timely as it improves our understanding of warming responses of species from different elevation origins and how this may influence the community composition in montane areas. The study is well conducted, have an overall good design and the manuscript is well written. However, there are a number of flaws and missing information that will hinder full understanding and interpretation of the results in the current version of the manuscript. I therefore recommend revision before the paper can be accepted. The issues are:

1. The species selection is well justified based on their thermal preferences. However, the authors have overlooked the potential intra-specific variations in thermal tolerance. In general, provenances from different altitudes and latitudes may exhibit distinct thermal ranges. For instance, the montane species' germplasm was gathered from elevations ranging between 1900-2500 m a.s.l. with potentially 3-4°C temperature range. These differences can have influenced the thermal ranges of the collected species. Therefore, I recommend including details about the specific altitudes from which each species was collected and discussing potential effects of provenance on the results.

Response: In response to this query, we added the following under the experimental set up of the methods section in lines 276-283:

Specifically, seeds from montane species were collected from elevations ranging between 2200–2500 masl, within temperatures close to their T_{opt} , while lowland species were collected from elevations between 1300–2200 masl, within temperatures close to the cold portion of their geographic range. All seeds were collected from a minimum of three to a maximum of five trees per species to minimise intra-specific variation. All seeds were propagated in poly-pots in a nursery located at a site with mean annual temperature of 22°C and a minimum of 100 seedlings per species were produced.

2. One notable advantage of this experiment is the considerable effort dedicated to using uniform planting soil across all sites. Soil from a 0.32 m² pit at each tree was replaced with soil from the upper site and distributed to the other sites. However, a question arises regarding

how long the roots will remain within the replaced soil space. It is plausible that a transition period within the experimental three years will occur during which the root system gradually acclimates to the original soil conditions at the various sites.

Response: It is indeed possible that the roots of some trees have extended and are beyond the 80cm x 80cm x 50cm soil pit where the trees were initially planted. Nevertheless, there were no interactions between roots of different trees during the sapling establishment, reducing the competitive effect on survival. Additionally, native soils at all experimental sites have similar N, P, and pH (Table 2) with soil P varying between ~10 -29 Kgha⁻¹ and soil N varying from ~201-231 Kgha⁻¹ and pH ranging from 5.1 to 5.3. Therefore, we expect that the minimal variation across native soils is not the leading cause of observed growth differences across sites. In recognition of this we have added the following text in the discussion lines 442-444 as follows:

Although soil conditions were controlled in our experiment, root expansion to local soils is expected over time, and the slightly differing soil nutrients may influence tree performance once trees reach a suitable size⁵¹.

3. The soil quality to which the trees are adapted could also influence their performance in the soil from the upper sites, raising another question about potential differences in soil quality between sites, which preferably should be reported in the supplementary.

Response: We selected soils from a non-forest site nearby location to the 14°C MAT site, which is located next to the forest where the seeds were originally collected from. This alternative soil has a texture and nutrient profile similar to that of the forest soils where the source trees naturally grow, however some nutrients are lower (N, K) while others are similar (P, Ca, Mg) and slightly less acid and lower organic matter on the non-forest soil. Detailed information of soil nutrients and texture for both types of soil are provided in Supplementary Tables 1 and 2 and information from native soils at experimental sites is provided in Table 2 of the main text. Experimental sites tend to have lower N and P (except at the 22°C site) and higher pH than the forest soils. Our comparisons are done for trees growing under common soils. We do not know what would have been the exact effect of having the trees growing on the original forest soils and it was unrealistic to use those soils for planting under our experimental set up.

4. The application of additional fertilizers in one of the six blocks is a significant assessment of soil factors. However, especially as the authors did not detect a significant treatment effect, clarification is needed on the criteria used for selecting the quantity and quality of fertilizers, as well as the sensitivity of the significance test employed.

Response: We have added supplementary notes 1 with criteria used for selecting fertilisers. We assessed the amount of fertiliser to apply by comparing native soils and soils to use for planting with values from a reference soil of intermediate fertility. The fertilisation treatment aimed to remove nutrient deficiency and maintain intermediate levels of soil fertility following the reference soil values. The type and quantity of fertilizers were selected

according to conventional methods for tropical species nurseries and the team's own experience. Below is part of the text in Supplementary notes 1

The fertilization strategy aimed to remove nutrient deficiencies on fertilised trees using as reference values from intermediate fertility soils. Therefore, we aimed to increase P, K, Mg and pH while maintaining N and Ca (Supplementary Tables 1-3). Very small but frequent additions were applied to maintain moderate nutrient input (Supplementary Tables 3-6), as neotropical montane rainforests and tropical tree species typically respond rapidly to moderate nutrient additions, which influences productivity and growth^{1,5}.

Regarding the sensitivity of the significance test employed (to statistically test for nutrient effects) to the number of samples used: we added the following text on lines 341-342 showing the results of the test demonstrating there is no impact of sample size on the results obtained:

We employed a Welch ANOVA to account for differences in sample sizes between fertilised and non-fertilised trees and found there is no impact of sample size ($F = 3.24$, $p = 0.21$).

5. Tree growth was assessed by measuring stem diameter at 2 cm height, from which relative growth rates (RGR) were calculated and standardized within species by dividing each individual's RGR by the maximum RGR observed within that species. However, several key details that are crucial for understanding and interpreting the results are missing. For example, the frequency of diameter measurements and how stem irregularities at such a low height (2 cm) were addressed are not specified.

Response: As originally mentioned in the methods in the section 'Growth and survival monitoring', these measurements were taken on a quarterly basis. We have modified the original text to the following in lines 297-298:

Measurements were taken every four months and data from nine measurement campaigns were used in this study⁴¹

Regarding stem irregularities we added the following in lines 288-290

All trees were tagged, mapped, and their diameter at 2cm height was marked to ensure all posterior measurements were taken at the same stem position. If there were any stem irregularities, the 2 cm mark was moved to a nearby location.

6. Moreover, RGR typically decreases with size due to self-shading, canopy closure, and other factors. Therefore, I recommend including size data in the supplementary materials, along with growth rate trends over time if available. Since mortality rates may also vary with tree size, there is an additional reason to provide data on the size distribution of the trees and analyse the data in relation to potential differences in size.

Response: The data set used in this manuscript including size data per tree per species per site is found under reference 41 and is now cited in the manuscript as shown in the previous answer. We have also added Supplementary Table 7 which includes mean tree diameter, canopy size and height per species per experimental site before planting and after 3 years of

planting. Growth rate trends for each species at each experimental site over time are now provided in Supplementary Figure 4.

7. Other factors that could have influenced both mortality and growth include diurnal and seasonal variations in climate, such as day and night temperatures, VPD, and radiation.

Response: Variations in VPD, direct and diffuse PAR are shown in Table 2. There are differences in mean sites and maximum VPD across sites and to a lower extent in direct and diffuse PAR. Temperature variations across months is similar across sites and the overall variation in temperature across the year at individual sites is low (not shown), therefore we do not expect a seasonality effect that can interfere with the observed growth responses. We added the following caveat for factors such as VPD, solar radiation and cloud cover that were not possible to control in our experiment in lines 437-442 as follows:

Under the natural settings of a tropical elevation gradient is also not possible to control for i) indirect temperature effects via vapor pressure deficit which affects photosynthesis and water use efficiency and ii) incident radiation due to variations in cloud cover which influences tree growth and mortality. Nevertheless, the assumption of annual temperature as the main driver of species performance alongside elevation in a high precipitation region is still valid⁴⁹.

Also, increases in plant respiration can be an important source of mortality, we included the following about respiration which is also acclimating to warming in lines 500-506:

Furthermore, tree growth is the outcome of photosynthesis (carbon gains) minus plant respiration (carbon losses) and both processes have been shown to moderately acclimate to warming in tropical montane species^{42,44,57}. If both photosynthesis and respiration acclimate to changes in temperature, but tree growth is reduced, plants might be investing the net carbon gains in additional metabolites to thermoregulate, increase thermal tolerance or increasing root biomass to increase water uptake, but these hypotheses need to be tested.

8. Despite regular irrigation, precipitation may also have had an impact, but specific data on this is currently unavailable.

Response: Mean annual precipitation (2774, 2045, 2298 mm year⁻¹) at each site (14°C, 22°C, 26°C) MAT respectively was provided in the original submission in the methods section under study area and now is also included in Table 2 with other climatic variables measured by a meteorological station at each experimental location during the period Oct 2019- Jan 2022. Dry season duration at these sites is short with maximum number of consecutive days without rain (12.5, 21.4, 20.3 for the 14°C, 22° and 26°C MAT sites respectively).

9. Furthermore, it remains unclear whether the temperature ranges shown in Figure 1 encompass both day and night temperatures, or solely daytime temperatures. It is surprising that more detailed climate data are missing.

Response: The initially submitted Fig. 1 included only diurnal variation in experimental sites MAT, we have now modified it to include both day and nighttime. Fig. 1 caption includes the following:

Coloured vertical polygons represent the thermal environment measured at each experimental site with MAT (represented with vertical coloured lines) of 14°C, 22 °C, and 26°C with the lower and upper thermal limits represented by the 10th and 90th percentiles of MAT respectively. Note that the 90th percentile of the 22°C site coincides with the MAT of the 26°C site.

Table 2 provides now more information on climate data at each experimental site

10. The manuscript is lacking in citations of other relevant studies, particularly those involving common garden and growth experiments assessing temperature effects. While it does reference broader ecosystem-level studies, it misses comparisons with species-specific research. Given that this study connects species-level findings to ecosystem dynamics, its results should also be discussed in relation to such studies.

Response: In response to this we have added the following in the discussion lines 486-500 as follows:

Large species variability across experimentally applied temperature treatments in transplant studies has been observed, with effects ranging from positive to negative on survival, photosynthesis, growth and leaf functional traits^{19,47,54,55} supporting results obtained in this study. Such species differences may be related to different growth strategies⁵⁶. For example, significant variation in sensitivity to warming was found in a transplant experiment with Afromontane forest tree species in Rwanda. The survival of all studied species declined with rising MAT. However after establishment, early successional species showed faster growth with a 2°C increase in MAT, while some late successional species exhibited either no response or reduced growth¹⁹. In a warming experiment involving boreal and temperate forest species, it was found that species growing close the hot extreme of their thermal ranges exhibited reductions in net photosynthesis and growth, whereas species growing closer to their cold extreme responded positively to warming⁵⁵. Similarly, a seedling transplant experiment with dominant canopy-forming treeline species in the southern tropical Andes revealed species-specific differences and contrasting responses in seedling survival to changes in MAT⁵⁴.

A few other more specific comments:

11. Abstract/Last line: This sentence seems partly redundant: Maybe revise to "increased homogeneity in community composition, resulting in reduced biodiversity".

Response: We modified the sentence following the suggestion.

12. Page 9 and elsewhere: A more handy and commonly used unit for RGR would be % year⁻¹.

Response: Thank you for your suggestion regarding the use of % year⁻¹ for RGR. After careful consideration, we decided to retain mm mm⁻¹ year⁻¹ as our unit of measurement as the results or interpretation would not change.

13. Page 12, line 6: Text refer to Fig. 5 which do not exist. Should be Fig 4?

Response: It is figure 4, this has been corrected.

14. Page 12/13: The authors estimate that 27% of the montane species may be lost by 2050. Except for the assumed increase of 3 °C in the Andes by 2050, what is this estimate based on and how was it estimated?

Response: We decided to remove this calculation to avoid criticism for an extrapolation made based on an individual study.

15. Page 13 and the bottom: The estimated relatively higher sensitivity of 14% of montane species compared to lowland species. How robust is this estimated difference? Although it is clear that the direction is different the magnitude of the response is not evident.

Response: We have removed this sentence as this is only valid in our experimental results based on significant differences in the obtained regression slopes for montane and lowland species in Fig. 4.

Response to Reviewer #3

We thank the Reviewer for their very useful feedback. We have numbered each of the comments and provided a response in grey text following each comment. Responses refer to the line numbers of the manuscript file with highlighted changes. We have also copied the respective manuscript text in the response below in most cases (when it was suitable). The highlighted text in grey corresponds to new text while non highlighted is text from the originally submitted manuscript.

In this manuscript the authors describe the results of a warming experiment they conducted for analyzing heat tolerance of tropical montane trees along an elevation gradient in the Colombian Andes under the current context of climate warming. The results show that lowland species have higher survival rates than montane species at higher temperatures and growth rates are higher for lowland trees at higher temperatures but higher for montane trees at lower temperatures. The authors also found results vary across species. In general, these results provide valuable insights into the species composition change occurring in tropical mountains as a response to global warming. I believe that environmental scientist could find this paper relevant. In particular, the experiment conducted is very valuable since it required a lot of work and effort from the research team and provides interesting results. There are, however, some points I would like to raise regarding the limitations of the experiment that are important to discuss.

1 First, water is a fundamental component of how species survive and growth under a changing climate and needs more discussion. Although it is clear that water limitation was controlled for in the experiment, it is unclear whether sites received different amounts of water once precipitation is considered and if this could have had an effect on growth and survival. Ideally, the experiment could have included a control treatment and a water*temp treatment. Given the experiment is already complex as is and water was not considered as a

treatment, I suggest that the authors add some discussion on the possible effects of climate change including water, not only temperature.

Response: We chose to focus on a temperature gradient in an area where precipitation remained consistently high >2000 mm year⁻¹ across the different MAT in experimental locations with values originally written in the methods section and now also in Table 2. The maximum number of consecutive days without rain are 12.5, 21.4 and 20.3 at the 14°C, 22°C and 26°C MAT sites respectively with corresponding mean annual precipitation measured at each site during Oct 2019-Jan 2022 of 2274, 2045, 2298 mm yr⁻¹. Thus, the 14°C site with the largest precipitation receives less irrigation. To acknowledge the lack of water control at our experimental locations and its possible influence on tree performance we have added the following in the discussion lines 431-437 as follows:

Overall, our experimental findings highlight that rising temperatures pose a real threat to many tropical plants despite of not controlling for all possible factors that vary under natural montane forest settings. Although our experimental sites are located in areas with high precipitation (>2000 mm yr⁻¹) with relatively short rainy season (maximum ~ 21 consecutive days without rain), our study has limitations in understanding the effects of differing levels of water availability on survival and growth and the effects of precipitation variability in the Andes^{4,11-13}.

And in lines 450-451:

Since the experiment used controlled soil conditions and constant watering, the results may not fully represent the complex interactions present in natural, variable environments.

We modified an existing sentence regarding amount of water being added in lines 290-294 as follows:

All trees have been irrigated since planting, using an average of 8 to 24 litres of water per night when there were no rain events during two consecutive days to allow successful plant establishment and avoid effects of water limitation. See Table 2 for maximum number of consecutive days without rain at experimental sites.

We added a paragraph in the introduction to provide more context of the work including a justification to focus on only impacts of warming in lines 56-74 as follows:

The tropical Andes are among the most biodiverse regions in the world¹⁻³. However, its diversity is at risk due to climate change and habitat loss^{1,4}, which may affect their functioning and capacity to provide ecosystem services, key for the people in this and adjacent regions^{5,6}. The impacts of climate change are highly significant in the northern South America, where both monthly mean and maximum temperatures have increased between 0.6 and 2.4 °C and between 1.2 and 6.6°C respectively, during the 1950-2010^{4,7-9} period. Future projections in this region correspond to 4.5°C rise in the median temperature by 2100 (with 2.6 and 6.6 °C as the 5th and 95% percentiles) relative to present day following SSP5-8.5¹⁰. Similar future temperature projections are expected in other tropical montane regions such as in Central Africa but are ~ 1°C larger than projections for montane forest in central America. The expected temperature increase across the tropical Andes is elevation-dependent⁴ which amplifies the difference between minimum and maximum temperature both diurnally and annually, as well as the frequency of heatwaves^{4,11}. In contrast, observed annual

precipitation trends do not show a homogeneous pattern across the Andes during the period 1964-2008^{4,12}, varying between -4% and +4% per decade relative to mean annual precipitation. Large uncertainties remain in future precipitation projections over the Andean region^{4,11-13}. It is expected that the predicted systematic temperature rise will affect natural ecosystems across the tropical Andes however the extent and nature of the of the impacts remains understudied.

2. Second, authors should address phylogenetic, genetics and epigenetics effects on the results of the experiment. For example, all lowland species are genus *Inga*, could this represent any bias in the results? Also, seeds were collected from the same forest close to the 14 degrees C site instead of along the elevation gradient. Does this mean that seeds for lowland species were collected closer to the coldest extreme of their geographic range? Could this have an effect on the results?

Response: As explained in the text we selected 15 out of the 37 most dominant tree species in the Colombian Andes (40 species in total when accounting for three dominant palm species) most of which are of intermediate succession and are also sun germinating species. Also explained in the methods: our selection criteria included species with high and low T_{opt} . Not in explicit in the text but implicit to the study, our high temperature threshold of T_{opt} as the mean annual temperature at the hottest site i.e 26°C. Detail not explicit in the text: most of the dominant species of intermediate succession (with easy to germinate seeds) with high T_{opt} (> 20°C) and below 26°C belonged to the *Inga* genus. There were two other two suitable non-*Inga* species *Saurauia laevigata* and *Alchornea triplinervia* with seeds that are not easy to germinate. Instead, *Inga* seeds are easy to germinate with high survival rates. For this reason, we selected species from the *Inga* genus.

Phylogenetic differences do not affect the species level survival and growth analysis presented in figures 2 and 3. In response to this, we added a mixed effect model to relate TDI and SGR (Fig 4) in which we included the botanical family as a random factor to control for the taxonomic bias in our experimental setup. This is explained in lines 344-352 as follows:

To evaluate the relationship between growth rate (SGR) and the thermal displacement index (TDI), where each observation represents one species per site, we employed a mixed-effect linear model. We also added species group as a covariate (fixed factor) to assess for differences in species response. We run a model for TDI with each of the three MAT metrics (MAT, MAT^{90th} and MAT^{10th} percentiles). Additionally, we used the botanical family as a random factor to control for taxonomic bias in our sampling, where all lowland species belong to genus *Inga* (Fabaceae), and three montane species belong to the Clusiaceae family. We reported conditional (fixed plus random effects) and marginal (only fixed effect) R^2 and compare both to detect the effect of taxonomic bias in our results.

We found that the effect of taxonomic bias is lower in the difference between conditional and marginal R^2 of the mixed model suggesting that our results are not statistically biased. This is in lines 401- 403 as follows:

Furthermore, we did not find large differences between conditional ($R^2 = 0.57$) and marginal ($R^2 = 0.52$) effects after accounting for the taxonomic bias in our results.

We also added in the discussion the following lines 451-455:

Our study includes 15 of the 37 (40%) most dominant species from the Colombian Andes which belong to 8 of the 13 dominant taxa, implying that although our experiment has a high representation of dominant taxa (61%), species responses may be shared among close relatives.

3. Third, the authors claim their results provide insights into interspecific competition but this was not tested in the experiment (there aren't treatments for testing interspecific competition in the experiment). Authors should mention how interspecific interactions could influence their results and tone down their conclusions on competition. This is important since the manuscript revolves around the concept of thermophilisation.

Response: As explained in the methods, trees were planted in open areas 2.5 m apart to avoid competition during the first 3 to 4 years of growth at experimental sites. This has been clarified in the methods section under experimental set up with the following lines 269-271:

Each block contains one tree of each of the 15-study species, planted 2.5 m apart from each other to avoid competition during the first three to four years of growth at experimental locations.

Supplementary Table 7 includes the average size of each species after three years of planting and before planting. At experimental sites with 14°C and 22°C MAT, maximum stem height was 165 cm and 310 cm respectively and maximum canopy diameters were 97 cm and 213 cm respectively. These measurements and the distance between trees (2.5m) mean that the individual crowns are distant, avoiding competition for light between trees. We also expect that roots do not extend longer than their own crown diameter (*Enquist 2022, tree Physiol., 22(15-16): 1045*). Therefore, we do not expect competition between trees at the 14°C and 22°C MAT sites during the study period. At the 26°C MAT site, only trees from lowland species have survived, with heights varying from ~300 up to 426 cm with maximum canopy diameter varying between 212 and 452 cm. We expect that at this site, there will be competition between the four lowland species. However, our question in this research is whether growth and survival of both montane and lowland species differ when growing under the warm extreme of the montane species (22°C MAT site) and under the cold portion and cold thermal extreme of lowland species (14°C MAT site).

Since we do not expect competition at the 14°C and 22°C MAT sites, we use the obtained growth rates as proxy for interspecific competition abilities assuming that a fast growth rate implies advantage over other species in terms of resource acquisition.

We modified the text to tone down the interpretation of our findings in lines 474-481 as follows:

The observed pattern of thermophilisation across Andean forests shows a heterogeneous trend, with rapid changes in tree composition at some sites but slower rates at others; in certain locations, montane tree abundance has even been observed to increase^{14,15}. Although this experiment did not directly but indirectly evaluate interspecific competition effects, our results could provide a possible explanation for this pattern of heterogeneity: the replacement rate of montane species which die, i.e. those unable to tolerate increased warming, with lowland species is affected by the presence of the dominant heat-tolerant montane species, which can maintain or increase their abundance under moderate warming.

4. Finally, topographic effects are not mentioned in the manuscript but are very important in mountains. High topographic complexity provides a wide climatic space and could be relevant for species persistence. Please add some discussion on this topic as it could be playing a role in the variation of thermophilisation and it is related to the claim the authors make about using coarse temperature data to study community composition change. 4

Response: We tried minimised topographic variation by selecting sites with similar topography, however we do not have specific measurements of topography. This was added in lines 253-254 as follows:

Sites with similar topography were selected to minimize its influence on plant performance.

And in discussion lines 444-447:

High topographic complexity can provide a diverse climatic space, which is crucial for species persistence. Experimental sites were selected with similar topography to reduce differing topographic effects on tree growth⁵¹.

Besides addressing these points throughout the manuscript, I suggest the authors make the following changes in specific sections and lines:

5. In the introduction, authors should provide a working hypothesis and predictions.

Response: In the introduction we added the following lines (111-122) on expectations of tree performance under the hot and cold extreme of species thermal distributions

According to Shelford's law²³, plant performance is limited by any deficit or excess of environmental conditions or resources (e.g., temperature), leading to a gradual reduction in performance from optimal environment conditions at which a plant grows best to extreme conditions under which a plants performs poorly^{24,25}. However, responses of plant performance to extreme cold and to extreme high temperatures may differ due to contrasting metabolic constraints (e.g., chilling vs. heating). The lower limit of a species' thermal distribution is considered a strong limiting factor to plant performance because cool conditions reduce metabolic rates²⁶, and thus growth. Conversely, the upper range and hot extreme of a species' thermal distribution may abruptly restrict enzymatic activity^{27,28}. Therefore, the displacement of a species from its thermal optimum towards the species' hot extreme could thus lead to a steeper reduction in tree growth compared to species displacement towards their cold extreme.

We added hypotheses following text on what the experiment will test in lines 150-158:

To further our understanding of the mechanisms underpinning forest compositional change, here we tested 1) whether dominant montane species are able to survive and grow at their hot extreme and beyond this hot thermal limit, 2) whether growth and survival of dominant lowland species is higher than for the dominant montane counterparts under the hot extreme of montane species thermal range and 3) under the cold extreme of lowland species thermal range. We hypothesised that survival and growth of all species would decrease when growing away from their thermal optimum and that plants growing under the cold extreme of their thermal range would have better performance in terms of growth and survival than plants growing under their hot extreme.

6. Lines 85 to 105 are Methods. Authors should make this paragraph shorter and move detailed info to Methods.

Response: Following this comment, we moved a lot of material that was in the introduction to the methods section and kept what we considered relevant material for the introduction.

7. The introduction should include more information about what's the expected temp change for this tropical mountain range and put into global context this specific case (what's happening in other tropical mountains?).

Response: We added the following text in lines 61-72 including temperature and precipitation projections

Future projections in this region correspond to 4.5°C rise in the median temperature by 2100 (with 2.6 and 6.6 °C as the 5th and 95% percentiles) relative to present day following SSP5-8.5¹⁰ Similar future temperature projections are expected in other tropical montane regions such as in Central Africa but are ~ 1°C larger than projections for montane forest in central America. The expected temperature increase across the tropical Andes is elevation-dependent⁴ which amplifies the difference between minimum and maximum temperature both diurnally and annually, as well as the frequency of heatwaves^{4,11}. In contrast, observed annual precipitation trends do not show a homogeneous pattern across the Andes during the period 1964-2008^{4,12}, varying between -4% and +4% per decade relative to mean annual precipitation. Large uncertainties remain in future precipitation projections over the Andean region^{4,11-13}.

8. Add more context to your study in the introduction. How are montane tree communities changing around the tropics?

Response: The second paragraph of the introduction explains this now in lines 75-110 as follows:

There is evidence of strong effects of global warming on tropical tree species, including shifts in their geographical range with consequences for population stability and community composition¹⁴⁻¹⁷. Observed shifts in tropical tree community composition are consistent with the idea that trees respond to changes in temperature by tracking the range of environmental temperatures within which a species can survive i.e. tracking their thermal range^{17,18}. A species' thermal distribution comprises the range of temperatures at which the species is found; within this range, the temperature at which the species grows best is known as their thermal optimum, T_{opt} (Fig. 1). We define the cold and warm portions of a species thermal range as the variation in minimum and maximum temperatures experienced by the species, respectively. The hot extreme of a species thermal range can be defined as the temperature above the 75th percentile of the maximum temperature experienced by the species, and their cold extreme as the minimum temperature below the 25th percentile of the minimum temperature experienced by the species. The upward movement of species in montane environments from the warm lowlands to cooler uplands produces a reconfiguration of species communities which has been termed thermophilisation¹⁵: warm affiliated lowland foothill thermophilic species, hereafter termed lowland species, are increasing in abundance across elevations relative to highland cold affiliated montane species, hereafter termed montane species. Observed directional shifts in species composition over time detected on forest plots^{14,15,19,20} provide evidence of thermophilisation on tropical montane tree communities in the Andes (reported

in Colombia, Peru, Ecuador and Argentina^{14,15}), in Afromontane forests (reported in Rwanda, Uganda, Democratic Republic of Congo and Tanzania²⁰) and in Central America (reported in Costa Rica²¹ and Jamaica²²). Thermophilisation in Andean forest is consistent with concurrent warming in the region and is caused by 1) increased abundance of lowland species in their upper limit of elevational range which coincides with the cold extreme of the thermal range, expanding their elevational range^{14,15,20} and 2) increased mortality of montane species in their lower limit of their elevational ranges (i.e. the range of elevations within which species can survive) which coincide with the hot extremes of their thermal ranges, leading to contractions of their elevational range¹⁹. However, to date it is not known whether the loss of montane species is due to direct negative impacts of climate warming on tree growth, or whether losses are driven predominantly by increased competition with lowland species. Such understanding is crucial for predicting future rates of change, as well as for planning conservation programmes. Furthermore, although observed species compositional shifts in the tropical Andes support thermophilisation, such change in species composition is heterogeneous across elevations^{14,15}. This is expected to be due to differences in species level responses global warming^{14,19}. Therefore, understanding species-level responses will enhance our knowledge of the resilience of tropical montane ecosystems to global warming.

9. In Methods, the authors should explain whether the data used for estimating thermal ranges per species encompass the whole geographic distribution area of each spp or if it is only based on the presence of species within the area of study. It would be useful to add the number of records used per species to get these thermal ranges, you could summarize this information on Table S1 and add it to the main text and Figure S2.

Response: data to estimate thermal ranges is based on species presence (as reported) within the whole geographic distribution area of each species. This is now clarified in lines 177-184:

For each of the species (selected from the Colombian Andes data set), we estimated their geographical thermal range based on biological records and global interpolated climatic data from the whole of the Andes region which encompasses the whole geographic distribution area of each species. We used 6958 spatially unique observed species presence records at 30 arc seconds (~1km) reported in the botanical information and ecology network, BIEN³⁷) which provides access to a number of different public datasets including the Global Biodiversity Information Facility (GBIF)³⁷. To avoid multiple clustered observations, only one record per grid cell was used

The number of records used to estimate species level thermal distributions is included in Table 1.

10. Seeds were collected close to an elevation which MAT is 14 degrees C. Why were seeds collected close to the 14°C and not across the study area?

Response: We collected seeds near the 14°C experimental site, which is located next to a large extension of montane forest. The forest remnants are scarce across the tropical Andes and is difficult to obtain large amounts of seeds from nurseries for many native and dominant tree species. For this reason, we limited seed sampling for our experiment to one large forest fragment that covers an elevational range from 1300-2500masl. A high proportion of seeds were sampled at elevations above 2000masl due to frequency of species. Also, we limited our sampling to few individuals per species to reduce intraspecific variability. We have

clarified the altitude ranges for each species group (montane and lowland) in the Methods section in lines 276-283 as follows:

Specifically, seeds from montane species were collected from elevations ranging between 2200–2500 masl, within temperatures close to their T_{opt} , while lowland species were collected from elevations between 1300–2200 masl, within temperatures close to the cold portion of their geographic range. All seeds were collected from a minimum of three to a maximum of five trees per species to minimise intra-specific variation. All seeds were propagated in poly-pots in a nursery located at a site with mean annual temperature of 22°C and a minimum of 100 seedlings per species were produced.

11. Did you also measure how much water plots received by precipitation?

Response: There is a meteorological station at each experimental site and annual precipitation values (2274, 2045, 2298 mm yr⁻¹ at the 14°C, 22°C and 26°C MAT sites respectively) were originally reported in the submitted manuscript. A summary of mean climate variables recorded during the period October 1st, 2019, Jan 31st, 2022, is now included in Table 2.

12. Are there any phylogenetic trends in tree growth that should be considered here? For example, all four lowland spp are Inga. Do Inga trees have faster growing rates in general?

Response: Phylogenetic differences do not affect the species level survival and growth analysis presented in figures 2 and 3. The Inga genus has a faster growth rate relative to other neotropical genus. In response to this, we added a mixed effect model to relate TDI and SGR (Fig 4) in which we included the botanical family as a random factor to control for the taxonomic bias in our experimental setup. This is explained in lines 344-352 as follows:

To evaluate the relationship between growth rate (SGR) and the thermal displacement index (TDI), where each observation represents one species per site, we employed a mixed-effect linear model. We also added species group as a covariate (fixed factor) to assess for differences in species response. We run a model for TDI with each of the three MAT metrics (MAT, MAT^{90th} and MAT^{10th} percentiles). Additionally, we used the botanical family as a random factor to control for taxonomic bias in our sampling, where all lowland species belong to genus Inga (Fabaceae), and three montane species belong to the Clusiaceae family. We reported conditional (fixed plus random effects) and marginal (only fixed effect) R^2 and compare both to detect the effect of taxonomic bias in our results.

We found that the effect of taxonomic bias is lower in the difference between conditional and marginal R^2 of the mixed model suggesting that there is no significant taxonomic bias in our results. This is in lines 401-403 as follows:

Furthermore, we did not find large differences between conditional ($R^2 = 0.57$) and marginal ($R^2 = 0.52$) effects after accounting for the taxonomic bias in our results.

We also added in the discussion the following lines (451-455):

Our study includes 15 of the 37 (40%) most dominant species from the Colombian Andes which belong to 8 of the 13 dominant taxa, implying that although our experiment has a high representation of dominant taxa (61%), species responses may be shared among close relatives.

13. L.63 Define thermophilisation.

Response: A definition has been added in lines 87-92 as follows:

The upward movement of species in montane environments from the warm lowlands to cooler uplands produces a reconfiguration of species communities which has been termed thermophilisation¹⁵: warm affiliated lowland foothill thermophilic species, hereafter termed lowland species, are increasing in abundance across elevations relative to highland cold affiliated montane species, hereafter termed montane species.

14 L72 Unclear what ‘these conditions’ are, what conditions?

Response: The sentence now reads as follows in lines 150-155:

To further our understanding of the mechanisms underpinning forest compositional change, here we tested 1) whether dominant montane species are able to survive and grow at their hot extreme and beyond this hot thermal limit, 2) whether growth and survival of dominant lowland species is higher than for the dominant montane counterparts under the hot extreme of montane species thermal range and 3) under the cold extreme of lowland species thermal range.

15 L72-73 In ‘We further tested the performance of lowland species at high elevations’ explain temp component linked to elevation. High elevation = low temperature.

Response: we modified the sentence and now reads as follows in lines 155-158:

We hypothesised that survival and growth of all species would decrease when growing away from their thermal optimum and that plants growing under the cold extreme of their thermal range would have better performance in terms of growth and survival than plants growing under their hot extreme.

16. L.76 ‘lowland trees species’ == ‘lowland tree species’ (remove s in trees)

Response: We have gone through the text and corrected this.

17. L.95 My understanding is that 26 degrees C is outside of all spp temp range (including lowland spp) not close to the T_{opt} of lowland trees

Response: While 26 degrees Celsius is indeed outside the temperature range for montane species, it is close to the optimum for two lowland species, as shown in the original figure 1, modified figure 1 (T_{opt} denoted in red and blue dots for lowland and montane species respectively) and Table 1.

18. L.98 Could you provide the approximate temperature that is expected in the future in these mountains? To contextualize 26 degrees C

Response: 26°C is close to the T_{opt} of two of the lowland species (Table 2) and provides a warm treatment for the other two lowland species with T_{opt} at 22°C MAT and is the hot extreme of the montane species. Based on future projections for SSP5-8.5 with reference to present day, by end of the century of a 4.5°C rise in the median temperature (with 2.6 and

6.6 °C as the 5th and 95th % percentiles), temperatures at the current 22°C site could rise to be as the ones in the current 26°C site.

The following text on future temperature projections have been added in the first paragraph of the introduction in lines 61-66:

Future projections in this region correspond to 4.5°C rise in the median temperature by 2100 (with 2.6 and 6.6 °C as the 5th and 95% percentiles) relative to present day following SSP5-8.5¹⁰ Similar future temperature projections are expected in other tropical montane regions such as in Central Africa but are ~ 1°C larger than projections for montane forest in central America.

19. L.99 ‘closet’ == ‘close’ (remove t)

Response: we have gone through the text and have corrected it.

20. L.101 From figure 1, it doesn't look like 22 degrees C is close to the coldest end of any of the lowland tree thermal ranges. All four spp reach the blue box in Figure 1 suggesting some individuals already experience average temp close to the mean temp in the colder site

Response: We modified Fig 1 to include variation within the minimum and maximum temperatures experienced by each species based on species records and temperatures from the WorldClim V.2. dataset¹⁹ for the 1970-2000 period. The legend explains:

Boxplots show the variation in the minimum temperature, T_{min} (dark grey boxes) and the maximum temperature T_{max} (light grey boxes) at locations where species were recorded, representing the cold and warm portion for each species thermal range. T_{min} and T_{max} correspond to the average temperature of the coldest and warmest month respectively during the 1970-2000 period from the WorldClim V.2. dataset¹⁹ for each species record. The cold and hot extremes of the thermal distribution of each species correspond to temperatures below the 10th percentile of T_{min} and temperatures above the 90th percentile of T_{max} respectively

For the four lowland species, part of the variation in minimum temperatures (in dark grey boxes) overlaps with variation at the 22°C site (in light blue).

21. L.124 Include years of climate normal (years used in worldclim to estimate MAT, T_{min}, and T_{max})

Response: This is now included lines 188-193 as follows:

From locations where species occurrences were registered, for each species occurrence, we recorded the mean annual temperature (MAT), the minimum temperature (T_{min}) defined as the average minimum temperature of the coldest month and the maximum temperature (T_{max}) defined as the average maximum temperature of the warmest month (T_{max}) from the WorldClim V.2. dataset³⁹ reported for the period 1970-2000.

22. L.118-130. This paragraph has several typos, please revise

Response: we have gone through the text and corrected it.

23 L.132 Andean spp are distributed in a wide elevation range and some spp probably also on a wide geographic range? Please clarify this (it's unclear if you're considering the whole geographic distribution area of each spp or only data of the study area).

Response: We used the whole geographic distribution of species to extract thermal ranges. It was clarified in the method section included lines 178-184 as follows:

we estimated their geographical thermal range based on biological records and global interpolated climatic data from the whole of the Andes region which encompasses the whole geographic distribution area of each species. We used 6958 spatially unique observed species presence records at 30 arc seconds (~ 1km) reported in the botanical information and ecology network, BIEN³⁷ which provides access to a number of different public datasets including the Global Biodiversity Information Facility (GBIF)³⁷. To avoid multiple clustered observations, only one record per grid cell was used.

Also, we have included the number of records used per species in Table 1.

24. L.131-145 Is this the reason behind sites selection? If so, please make it explicit here.

Response: The classification of lowland and montane species was established prior to species selection and served as a criteria for species selection for the experiment. Site selection was primarily based on temperature gradients (with similar precipitation patterns in areas that facilitated logistical processes, such as access, planting and data collection) that allowed hypothesis testing regarding impacts of warming and cooling on montane and lowland species mimicking thermophilisation.

The following text was added in lines 234-237:

Experimental locations were selected based on i) thermal ranges of the selected species, i.e. sites with mean annual temperatures that allowed hypothesis testing and on ii) logistical constraints including access to private land with owner's approval for planting and to collect data.

25 L.134. The k-means was computed using temp and elevation data of all records of the 40 dominant spp within the study area, correct? Or was it using all temp and elevation data space within the study area (all pixels)? Please clarify. Also explain what low and high elevation mean in this context (for example, is low between 500 and 1000 m asl?)

Response: The k-means was performed over the whole tropical Andes. We removed the text on low and high elevation to avoid confusion. We also modified the text to clarify what was done as follows in lines 204-210:

Andean species are distributed within a wide elevation/temperature range, therefore different growth responses to changes in temperature are expected¹⁴. Two groups of species were observed on the estimated species' thermal distributions with varying values of T_{opt} associated to low and high thermal environments. For this reason, we performed a cluster analysis (k-means) across temperature and elevation in the whole tropical Andes from 500 to 3500 masl to partition species' thermal space to identify a break point in terms of temperature and elevation between them.

26 L.154 'based on above criteria' == 'based on the above criteria' (add the)

Response: Thanks for pointing this out we have gone through the text and correct it. Line 224:

Based on the above criteria

27 L.154 You already mention you selected 15. Change to ‘we selected four lowland and 11 montane species’

Response: Thanks for the comment, we modified that in lines 224-225:

Based on the above criteria, we selected a total of four lowland and eleven montane species

28 L.155 what do you mean with thermal ranges between 9 and 15 degrees C, is this only for lowland trees, are 9 and 15 average T_{opt} ? According to Figure 1, the lower end of thermal ranges of some spp are lower than 9 and the higher end are way higher than 15.

Response: The thermal range was defined in the preceding lines to the original L155 now in line 221-222 and refers to the difference between the maximum (T_{max}) and minimum (T_{min}) temperatures experienced by the species, as detailed in Table 1. The values of 9 and 15 degrees Celsius are not average T_{opt} values but represent the variation in thermal ranges for the study species.

The text was modified to the following in lines 224-229:

Based on the above criteria, we selected a total of four lowland and eleven montane species with T_{opt} between 13°C and 25°C with thermal ranges (difference between average T_{max} and T_{min}) between 9°C and 15°C. The lowland group includes two species with T_{opt} close to 22°C, and two species with T_{opt} close to 26°C. The montane group consists of eleven species whose T_{opt} is close to 14°C with average T_{max} up to 22°C, except one species which average T_{max} goes up to 24°C (Fig. 1, Table 1).

29L.203-204 Only one equation is needed

Response: Thanks for the comment, we keep one equation. Line 305.

30. L.212 ‘relative growth rate from and individual’ == ‘relative growth rate from an individual’ (remove d)

Response: Thanks for pointing this out we have gone through the text and correct it. Line 313

Relative growth rate from an individual tree.

31. L. 227 What predictors did you use in the regressions? Please explain further.

Response: Cox regression analyse survival of species along time, using the number of the day as a dependent variable in the regression.

32. L.248 nine out of 11 montane spp survived where? At the 22 degrees site? Please clarify.

Response: The following text was added in lines 361-362:

However, species level responses varied, and nine out of eleven montane species survived at the 22°C site

33. L.261 check parenthesis

Response: Thanks for pointing this out we have gone through the text and correct it. Line 374-375:

Montane species decreased RGR in response to warming ($F=5.6$, $p<0.001$).

34. L.266 remove dash between four and lowland

Response: Thanks for pointing this out we have gone through the text and correct it. Line 380:

Of the four lowland species, two maintained the same RGR

35. L.267 with two of the growing temp you mean two sites? Unclear, please rephrase.

Response: The following text was added in lines 380 -381

Of the four lowland species, two maintained the same RGR when growing at 22° C and 26°C MAT (Fig. 3a-d).

36. L.274 Fig 3 does not show these results, is this a t-test between lowland and montane groups? Please explain here and in Methods

Response: This was an additional analysis performed to compare between groups at the 22°C site, to test '2) whether growth and survival of dominant lowland species is higher than for the dominant montane counterparts under the hot extreme of montane species thermal range' (lines 152 154).

The specific text in the results did not refer to figure 3. We add the following in the method section lines 342-343 as follows:

Also, we compared the growth rate of montane and lowland species at the 22°C site using a t-test.

37. L.275-280 You could make this paragraph shorter by removing repetitive information

Response: We modified these paragraphs in the lines 391-394:

We can predict the magnitude of the observed growth responses to changes in temperature for all species when we separate the lowland and montane groups. Specifically, we found a strong relationship between scaled growth rate (SGR) and thermal displacement index (TDI) which represents the thermal displacement from the species T_{opt} , (Fig 4).

38. L.275-284 Are these correlations? Linear regressions? Or what type of model you used for calculating these relationships? Add to Methods (perhaps in L.216). Also, correlation does not imply that you can predict a response, it only helps explain a pattern, please rephrase here and in next paragraph.

Response: We used a mixed-effect linear model to assess the relationship between TDI and SGR. But also reported Pearson's R in Fig 4 and new Supplementary Fig. 3. We added the following text in lines 344-352 of the methods section:

To evaluate the relationship between growth rate (SGR) and the thermal displacement index (TDI), where each observation represents one species per site, we employed a mixed-effect linear model. We also added species group as a covariate (fixed factor) to assess for differences in species response. We run a model for TDI with each of the three MAT metrics (MAT, MAT^{90th} and MAT^{10th} percentiles). Additionally, we used the botanical family as a random factor to control for taxonomic bias in our sampling, where all lowland species belong to genus *Inga* (Fabaceae), and three montane species belong to the Clusiaceae family. We reported conditional (fixed plus random effects) and marginal (only fixed effect) R^2 and compare both to detect the effect of taxonomic bias in our results.

39. L.299 45% of trees or 45% of species?

Response: Thanks for the comments, we clarify that point. The following text was added in lines 414-418:

Although 55% of montane trees did not survive at the extreme of their thermal ranges, 45% (from nine out of the eleven planted dominant montane species) did survive under these conditions: four species decreased growth and five species did not change growth when growing at their hot extreme, which is on average 6.5°C higher than their T_{opt} .

40. L.321 I don't think these results suggest previous colonization of higher lands in warmer periods since you didn't analyze any historical data

Response: We agree, our analysis does not include historical data. However, with current and recent rise in temperatures in the Andes, lowland species may have the potential to colonize higher elevations. For this reason, we suggest that during past warming periods, lowland trees may have been able to colonize high-elevation areas despite experiencing temperatures below their thermal optimum in the lines 457-461 as follows:

In the highlands, where montane species are abundant, lowland species such as the four lowland species from the same genera used in this study can survive, albeit with reduced growth rates. This could suggest that during prior periods of warming, lowland species may have been able to colonize high-elevation areas despite experiencing temperatures below their thermal optimum²⁹.

41. L.320-326 Please add references

Response: The following text and references were added in lines 457-473:

In the highlands, where montane species are abundant, lowland species such as the four lowland species from the same genera used in this study can survive, albeit with reduced growth rates. This could suggest that during prior periods of warming, lowland species may have been able to colonize high-elevation areas despite experiencing temperatures below their thermal optimum²⁹. However, their slow growth rates under these conditions likely prevent lowland species from outcompeting montane species. In our experiment, trees were growing in sun exposed open areas, where minimum temperatures are extreme with strong wind exposure mainly reducing lowland species performance⁵⁴ at the 14°C MAT experimental site. However, if trees would have been growing

under the more sheltered and less cool understory in highland forests, they may be able to grow faster than in open areas, as observed in forest plots^{14,15,20}. Thus, under future warming, lowland species may possibly compete with montane species in highland forest, reducing their abundance. It could be expected that lowland species form novel highland tree communities, potentially increasing local species richness due to the large number of species found in lowland forests¹⁸. However, the rate of species compositional shifts, the expansion of lowland species, and the spread of heat-tolerant montane species, as well as their consequences for tropical Andean communities, will likely depend on the pace of climate change¹⁰.

42. L.328-330 Please add references

Response: The original submitted text has the only two reported references on the heterogeneous trend of thermophilisation in the Andean forest. Therefore we made no change, only update the references numbers in lines 474-476 as follows:

The observed pattern of thermophilisation across Andean forests shows a heterogeneous trend, with rapid changes in tree composition at some sites but slower rates at others; in certain locations, montane tree abundance has even been observed to increase^{14,15}.

43. Add discussion on limitations of experiment

Response: We added the following in lines 431-455:

Overall, our experimental findings highlight that rising temperatures pose a real threat to many tropical plants despite of not controlling for all possible factors that vary under natural montane forest settings. Although our experimental sites are located in areas with high precipitation (>2000 mm yr⁻¹) with relatively short rainy season (maximum ~ 21 consecutive days without rain), our study has limitations in understanding the effects of differing levels of water availability on survival and growth and the effects of precipitation variability in the Andes^{4,11-13}. Under the natural settings of a tropical elevation gradient is also not possible to control for i) indirect temperature effects via vapor pressure deficit which affects photosynthesis and water use efficiency and ii) incident radiation due to variations in cloud cover which influences tree growth and mortality. Nevertheless, the assumption of annual temperature as the main driver of species performance alongside elevation in a high precipitation region is still valid⁴⁹. Although soil conditions were controlled in our experiment, root expansion to local soils is expected over time, and the slightly differing soil nutrients may influence tree performance once trees reach a suitable size⁵⁰. High topographic complexity can provide a diverse climatic space, which is crucial for species persistence. Experimental sites were selected with similar topography to reduce differing topographic effects on tree growth⁵¹. Furthermore, our study does not account for potential changes in biotic interactions due to warming or cooling, which could influence survival (e.g., herbivory³⁵), growth (e.g., mycorrhizal associations⁵²), and reproduction (e.g., pollination⁵³). Since the experiment used controlled soil conditions and constant watering, the results may not fully represent the complex interactions present in natural, variable environments. Our study includes 15 of the 37 (40%) most dominant species from the Colombian Andes which belong to 8 of the 13 dominant taxa, implying that although our experiment has a high representation of dominant taxa (61%), species responses may be shared among close relatives.

44. Figure 1 caption is unclear, please rephrase this part: “These include mean annual temperature (MAT), minimum temperature (Tmin), and maximum temperature (Tmax)

estimated from the WorldClim V.2. dataset¹⁹ from locations where occurrences registered for each species in the BIEN database.” Also, please explain what the box and whiskers represent and whether site’s temp range was also extracted from worldclim or measured on site (for how long?).

Response: Below is the text for the caption of the modified Figure 1

Fig 1 | Thermal distribution of study species and range of measured air temperatures at each experimental site during the period October 1st, 2019 to January 31st, 2022. Boxplots show the variation in the minimum temperature, T_{\min} (dark grey boxes) and the maximum temperature T_{\max} (light grey boxes) from locations where species were recorded in the BIEN database, representing the cold and warm portion for each species thermal range. T_{\min} and T_{\max} correspond to the average temperature of the coldest and warmest month respectively during the 1970-2000 period from the WorldClim V.2. dataset¹⁹ for each species record. The cold and hot extremes of the thermal distribution of each species correspond to temperatures below the 10th percentile of T_{\min} and temperatures above the 90th percentile of T_{\max} respectively. Blue and red dots represent T_{opt} of each species. Montane and lowland species are in blue and red fonts, respectively. To minimise temperature bias due to geographic occurrence errors, 5% of the data from each tail (5% and 95%) were removed. Coloured vertical polygons represent the thermal environment measured at each experimental site with MAT (represented with vertical coloured lines) of 14°C, 22 °C, and 26°C with the lower and upper thermal limits represented by the 10th and 90th percentiles of MAT respectively. Note that the 90th percentile of the 22°C site coincides with the MAT of the 26°C site.

45. In panel figures, please change the position of the letters (maybe next to the spp name). It is confusing to have the letters within the main plots. Also add what the letters mean in figure captions.

Panel letters corresponding to figures 2 and 3 are now outside the panels, next to the name of each species. We added to each figure caption the following text: Each panel corresponds to data for each of the study species.